# THE CONVERGENCE OF SECOND-ORDER SAMPLING METHODS FOR DIFFUSION MODELS

## ABSTRACT

Diffusion models have achieved great success in generating samples from complex distributions, notably in the domains of images and videos. Beyond the experimental success, theoretical insights into their performance have been illuminated, particularly concerning the convergence of diffusion models when applied with discretization methods such as Euler-Maruyama (EM) and Exponential Integrator (EI). This paper embarks on analyzing the convergence of the higher-order discretization method (SDE-DPM-2) under $L^2$-accurate score estimate. Our findings reveal that to attain $\tilde{O}(\epsilon_0^2)$ Kullback-Leibler (KL) divergence between the target and the sampled distributions, the sampling complexity - or the required number of discretization steps - for SDE-DPM-2 is $\tilde{O}(1/\epsilon_0)$, which is better than the currently known sample complexity of EI given by $\tilde{O}(1/\epsilon_0^2)$. We further extend our analysis to the Runge-Kutta-2 (RK-2) method, which demands a sampling complexity of $\tilde{O}(1/\epsilon_0^2)$, indicating that SDE-DPM-2 is more efficient than RK-2. Our study also demonstrates that the convergence of SDE-DPM-2 under Variance Exploding (VE) SDEs aligns with that of Variance Preserving (VP) SDEs, highlighting the adaptability of SDE-DPM-2 across various diffusion models frameworks.

## 1 INTRODUCTION

Diffusion models , also known as Score-based Generative Models (SGMs), are a powerful generative model which is widely used in image synthesis (Li et al., 2022; Rombach et al., 2022; Saharia et al., 2022), video generation (Harvey et al., 2022; Wu et al., 2023) and molecular design (Anand & Achim, 2022; Xu et al., 2022).

Diffusion models operate through two primary processes: the forward process and the backward (reverse) process. The forward process involves transforming the original data distribution into Gaussian noise via a Stochastic Differential Equation (SDE). During this process, the gradient of the log density function, known as the score function, is estimated by denoising score matching (Vincent, 2011) and sliced score matching (Song et al., 2020). The backward process, on the other hand, is capable of generating samples from the target distribution using the estimated score function. This is accomplished through an equivalent reverse SDE or a probability Ordinary Differential Equation (ODE). Specifically, the reverse SDE usually can generate more diverse and high-quality samples than the reverse ODE (Tachibana et al., 2021; Lu et al., 2022b), while the reverse ODE is faster than the reverse SDE (Li et al., 2024).

Anderson (1982) showed that a continuous backward process could converge to a target distribution using the ground truth score function. However, in real-world applications, we're often limited to estimating this score function based on the available data. Moreover, to implement the backward process in practice, we need to discretize the reverse SDE or reverse ODE.

Commonly used discretization methods include Denoising Diffusion Probabilistic Models (DDPM) (Ho et al., 2020), Denoising Diffusion Implicit Models (DDIM) (Song et al., 2021a), Exponential Integrator (EI) (Zhang & Chen, 2023) and DPM-2 (Lu et al., 2022a). Song et al. (2021b) illustrated that DDPM is essentially the Euler-Maruyama (EM) discretization of the reverse SDE, a first-order discretization method. The Euler-Maruyama method directly discretizes the reverse SDE's drift term, leading to a high discretization error. To address this issue, Lu et al. (2022a); Zhang & Chen (2023) designed a new discretization method named DPM or EI, which utilizes the linear part of

the drift term. Although DPM and EI are also first-order discretization methods, they can generate high-quality samples with fewer discretization steps than EM. Furthermore, Lu et al. (2022a;b) proposed second-order discretization methods, DPM-2 and SDE-DPM-2, which utilizes the linear part of the drift term and approximates the non-linear part using Taylor expansion the probability flow ODE and the reverse SDE, respectively. A similar second-order discretization method, Runge-Kutta-2 (RK-2), differs from SDE-DPM-2 in discretizing the linear part of the drift term in the reverse SDE. In the study by Lu et al. (2022b), SDE-DPM-2 is capable of producing high-quality samples with fewer discretization steps in comparison to both DDPM and SDE-EI. To avoid confusion, we refer to EI, DPM, and DPM-2 as the samplers for Probability ODEs and SDE-EI (SDE-DPM), SDE-DPM-2 as the sampler for reverse SDEs.

Several studies Yang & Wibisono (2022); Huang et al. (2024a); Li et al. (2024); Chen et al. (2024b) have investigated the convergence of the first-order and the second-order discretization methods the Probability Flow ODE, i.e. EI and DPM-2. However, the convergence analysis of reverse SDEs in Diffusion models remains somewhat unexplored. The significance of reverse SDEs is underscored by findings in Tachibana et al. (2021); Lu et al. (2022b), which demonstrate their superior performance over Probability ODEs in terms of sample diversity and quality. Recent studies, such as those by De Bortoli et al. (2021); Lee et al. (2023); De Bortoli (2022); Chen et al. (2023b;a), have primarily focused on the convergence properties of first-order discretization methods, including the Euler-Maruyama (EM) method and SDE-DPM. To achieve a KL divergence of $\tilde{O}(\epsilon_0^2)$ between the target distribution and the sampled distribution, SDE-DPM requires a sampling complexity of $\tilde{O}\left(\frac{1}{\epsilon_0^2}\right)$ (Chen et al., 2023a; Benton et al., 2024).

In contrast, a growing body of work Li et al. (2019; 2024); Wu et al. (2024); Chen et al. (2024a); Huang et al. (2024b) has investigated accelerated samplers for reverse SDEs. Notably, Li et al. (2024) proposed an acceleration algorithm using a variant of DDPM, and Wu et al. (2024) introduced a variant of the RK-2 method. Both methods achieve a KL divergence of $\tilde{O}(\epsilon_0^2)$ with an improved sampling complexity of $\tilde{O}\left(\frac{1}{\epsilon_0}\right)$. However, the convergence analysis of SDE-DPM-2 remains unexplored. Furthermore, experimental results indicate SDE-DPM-2 can generate samples with better FID score than the methods proposed in Li et al. (2024); Wu et al. (2024) with same discretization steps. This paper aims to address this gap by providing a convergence analysis of SDE-DPM-2.

OUR CONTRIBUTIONS

1. In our study, we are the first to investigate the sampling complexity of SDE-DPM-2. Our results demonstrate that for achieving a KL divergence of $\tilde{O}(\epsilon_0^2)$, SDE-DPM-2 requires a sampling complexity—or the necessary number of discretization steps—of $\tilde{O}(\frac{1}{\epsilon_0})$. This sampling complexity is notably more efficient than that of SDE-DPM method, which requires a complexity of $\tilde{O}(\frac{1}{\epsilon_0^2})$.

2. We further examine the sampling complexity associated with a different second-order discretization method, namely RK-2. This method demands a sampling complexity of $\tilde{O}(\frac{1}{\epsilon_0^2})$ which is worse than that of SDE-DPM-2 due to that RK-2 directly discretizes the linear part of the drift term in the reverse SDE, leading to a higher discretization error. Our analysis underscores the superior efficiency of SDE-DPM-2 over both EI and RK-2 in terms of sampling complexity.

3. We broaden our analysis to Variance Exploding (VE) SDEs, demonstrating that the convergence of SDE-DPM-2 under the VE-SDE framework aligns with that of Variance Preserving (VP) SDEs. This alignment underscores the adaptability of SDE-DPM-2 method across various diffusion models frameworks.

The following parts of this paper are organized as follows: Section 2 provides a brief overview of the preliminary concepts. Section 3 introduces the assumptions and the main results. Section 4 provides a sketch of the proof. Section 5 discusses the extension of our analysis to Variance Exploding (VE) SDEs. Finally, Section 7 concludes the paper with a discussion of the results and potential future research directions.

## 2 PRELIMINARY

[Song et al.](#) [(2021b)](#) delineates two principal types of forward processes: Variance Preserving (VP) SDE and Variance Exploding (VE) SDE. The VP-SDE maintains a bounded variance throughout its evolution, culminating in a distribution that resembles white noise, denoted as $\mathcal{N}(0, I_d)$. A distinguished example of VP-SDE is the DDPM, pioneered by [Ho et al.](#) [(2020)](#). In contrast, the VE-SDE is characterized by its variance which incrementally increases over time, a concept vividly illustrated through the Score Matching and Langevin Dynamics (SMLD) framework by [Song & Ermon](#) [(2019)](#). The focus of our discussion will be predominantly on the VP-SDE, owing to its widespread application in the theoretical exploration of diffusion models, as evidenced by the works of [Yang & Wibisono](#) [(2022)](#); [Chen et al.](#) [(2023a)](#); [Li et al.](#) [(2024)](#); [Chen et al.](#) [(2024a)](#). Furthermore, we will demonstrate the applicability of our findings to VE-SDE, broadening the scope of our analysis.

We will first review the forward and backward processes of diffusion models. Additionally, we will discuss the methods used to discretize the backward process, ensuring a comprehensive understanding.

### 2.1 FORWARD PROCESS

The forward process is defined as follows:
$$\mathrm{d}x_t = f(x_t, t)\mathrm{d}t + g(t)\mathrm{d}w_t, x_0 \sim p_0, t \in [0, T] \tag{1}$$
where $f : \mathbb{R}^d \times \mathbb{R} \to \mathbb{R}^d$ and $g : \mathbb{R} \to \mathbb{R}$ are the drift and diffusion coefficients, respectively. $w_t$ is a $d$-dimensional Brownian motion. The initial distribution of $x_0$ is $p_0$, which is the data distribution. We denote the solution of (1) at time $t$ as $x_t$ and use $p_t$ to denote the distribution of $x_t$. With the increment of time, the distribution of $x_t$ will converge to the white noise distribution $\mathcal{N}(0, I_d)$.

### 2.2 BACKWARD PROCESS

Suppose we run the forward process until time $T > 0$, ending at $p_T$. There exists a backward process ([Anderson](#), [1982](#)) which starts from $x_T \sim q_0 = p_T$, as follows (running backward from time $T$ to 0):
$$\mathrm{d}x_t = \left(f(x_t, t) - g(t)^2 \nabla \log p_t(x_t)\right) \mathrm{d}t + g(t)\mathrm{d}\tilde{w}_t$$
Where $\tilde{w}_t$ is a backward Brownian motion (with time flowing backward). The gradient of the logarithm of $p_t(x_t)$, $\nabla \log p_t(x_t)$, is the score function of $p_t(x_t)$. For convenience, we can rewrite the reverse process in forward time with $x_t^{\leftarrow}$ denoting $x_{T-t}$. Then the reverse process can be written as followed(from time 0 to $T$):
$$\mathrm{d}x_t^{\leftarrow} = \left(-f(x_t^{\leftarrow}, T - t) + g(T - t)^2 \nabla \log p_{T-t}(x_t^{\leftarrow})\right) \mathrm{d}t + g(T - t)\mathrm{d}w_t \tag{2}$$
We denote the distribution of $x_t^{\leftarrow}$ as $q_t$. [Anderson](#) [(1982)](#) showed that with $q_0 = p_t$, the marginal distribution of $x_t$ in the forward process (1) and $x_{T-t}^{\leftarrow}$ in the backward process (2) are the same:
$$p_t(x_t) = q_{T-t}(x_{T-t}^{\leftarrow})$$

**Backward Process with Estimated Score.** If we have access to $\nabla \log p_t(x_t)$ for all time steps $t$, we can run the backward process described in (2) to generate samples from the target distribution $p_0$. Nonetheless, acquiring the score function in real-world scenarios is often challenging. Consequently, we commonly resort to methods like denoising score matching ([Vincent](#), [2011](#)) and sliced score matching ([Song et al.](#), [2020](#)) to estimate it from data. We use the symbol $s(x_t, t)$ to represent the approximated score and substitute it into the backward process (2). Then, the backward process can be written as:
$$\mathrm{d}x_t^{\leftarrow} = \left(-f(x_t^{\leftarrow}, T - t) + g(T - t)^2 s(x_t^{\leftarrow}, T - t)\right) \mathrm{d}t + g(T - t)\mathrm{d}w_t \tag{3}$$

We adhere to the same settings used in the theoretical analysis of diffusion models ([Yang & Wibisono](#), [2022](#); [De Bortoli](#), [2022](#); [Chen et al.](#), [2023a](#)), where the function $f(x_t, t)$ is defined as $-x$, and $g(t)$ as $\sqrt{2}$. Consequently, the forward process, as described in equation (1), aligns with the Ornstein-Uhlenbeck (OU) process ([Maller et al.](#), [2009](#)). Within this framework, the distribution $x_t$ given $x_0$ is Gaussian with mean $e^{-t}x_0$ and variance $(1 - e^{-2t})I_d$:
$$x_t|x_0 \sim \mathcal{N}(e^{-t}x_0, (1 - e^{-2t})I) \tag{4}$$

Then the backward process (2) can be written as:
$$\mathrm{d}x_t^{\leftarrow} = (x_t^{\leftarrow} + 2s(x_t^{\leftarrow}, T - t)) \mathrm{d}t + \sqrt{2}\mathrm{d}w_t \tag{5}$$

## 2.3 DISCRETIZATION OF BACKWARD PROCESS

Given (5), we can implement SDE discretization methods to simulate the reverse process and generate samples from the target distribution $p_0$. Let $0 = t_0 \leq t_1 \leq \cdots \leq t_N = T$ be the discretization points. We denote the solution of (5) at time $t_k$ as $x_{t_k}^\leftarrow$, and use $x_k^\leftarrow$ to denote $x_{t_k}^\leftarrow$. We will introduce two discretization methods, EI and SDE-DPM-2, as representatives of first-order and second-order discretization methods, respectively.

**The EI scheme:** By discretizing the nonlinear term $s(x_t^\leftarrow, T - t)$ with $s(\hat{x}_k^\leftarrow, T - t_k)$, Then at each time interval $[t_k, t_{k+1}]$, we have

$$\mathrm{d}x_t^\leftarrow = (\hat{x}_t^\leftarrow + 2s(\hat{x}_k^\leftarrow, T - t_k)) \, \mathrm{d}t + \sqrt{2}\mathrm{d}W_t$$

By integrating the above equation, we have

$$\hat{x}_{k+1}^\leftarrow = e^{h_k}\hat{x}_k^\leftarrow + 2(e^{h_k} - 1)s(\hat{x}_k^\leftarrow, T - t_k) + \sqrt{e^{2h_k} - 1}z_k$$

where $h_k = t_{k+1} - t_k$ and $\hat{x}_k^\leftarrow$ is the solution for the EI scheme at time $t_k$. $z_k \sim \mathcal{N}(0, I_d)$ is the standard Gaussian noise.

**The SDE-DPM-2 scheme:** By discretizing the nonlinear term $s(x_t^\leftarrow, T - t)$ with $s_{T-t_k}(\hat{x}_k^\leftarrow) + s^{(1)}(\hat{x}_k^\leftarrow, T - t_k)(t - t_k)$, where $s^{(1)}(\hat{x}_k^\leftarrow, T - t_k)$ is the total first order derivative of $s_{T-t_k}(\hat{x}_k^\leftarrow)$ with respect to $t$. Then at each time interval $[t_k, t_{k+1}]$, we have

$$\mathrm{d}x_t^\leftarrow = \left(\hat{x}_t^\leftarrow + 2s(\hat{x}_k^\leftarrow, T - t_k) + 2s^{(1)}(\hat{x}_k^\leftarrow, T - t_k)(t - t_k)\right) \mathrm{d}t + \sqrt{2}\mathrm{d}W_t$$

By integrating the above equation, we have

$$\begin{aligned}\hat{x}_{k+1}^\leftarrow ={}& e^{h_k}\hat{x}_k^\leftarrow + 2(e^{h_k} - 1)s(\hat{x}_k^\leftarrow, T - t_k) + \sqrt{e^{2h_k} - 1}z_k \\ &+ 2(e^{h_k} - h_k - 1)s^{(1)}(\hat{x}_k^\leftarrow, T - t_k)\end{aligned}$$

As is introduced in Lu et al. (2022b), the total first-order derivative of $s(\hat{x}_k^\leftarrow, T - t_k)$ concerning $t$ is approximated with previous buffered values of $s(\hat{x}_{k-1}^\leftarrow, T - t_{k-1})$ and $s(\hat{x}_{k-2}^\leftarrow, T - t_{k-2})$, which does not require extra computation of the score function. We check its efficiency with the experiments on CIFAR-10 in empirical results in Section 6. The approximation is as follows:

$$s^{(1)}(\hat{x}_k^\leftarrow, T - t_k) \approx \frac{s(\hat{x}_k^\leftarrow, T - t_k) - s(\hat{x}_{k-1}^\leftarrow, T - t_{k-1})}{t_{k-1} - t_k} \tag{6}$$

Note that SDE-DPM-2, SDE-DPM-Solver-2M and SDE-DPM-Solver++(2M) in Lu et al. (2022b) are equivalent as they stated. The difference lies in the parameterization of the objective function where SDE-DPM-2 is based on $s_\theta$, SDE-DPM-Solver-2M is based on $\epsilon$-prediction objective and SDE-DPM-Solver++(2M) is based on data prediction model $x_\theta$. We focus on the denoising score matching form SDE-DPM-2 to maintain consistency with existing theoretical analysis work.

The key difference between EI and SDE-DPM-2 lies in how the score function is approximated within the update scheme at each step: EI scheme approximates the score function at time $[t_k, t_{k+1}]$ with $s_{T-t_k}(\hat{x}_k^\leftarrow)$, while SDE-DPM-2 scheme approximates the score function at time $[t_k, t_{k+1}]$ with $s_{T-t_k}(\hat{x}_k^\leftarrow) + s^{(1)}(\hat{x}_k^\leftarrow, T - t_k)(t - t_k)$. To ease the notations, we use $\frac{\partial s(\hat{x}_{t_k}^\leftarrow, t_k)}{\partial t_k}$ to denote the partial derivative concerning $t$ at time $t_k$, $\frac{\partial s(\hat{x}_{t_k}^\leftarrow, t)}{\partial t}|_{t=t_k}$, and $J_{s_t}$ to denote the Jacobian matrix of $s_t$.

We denote the distribution of $\hat{x}_k^\leftarrow$ as $\hat{q}_k$. Our goal is to bound the KL divergence between $p_0$ and $\hat{q}_T$, which will also yield a bound on the TV distance via Pinsker's inequality.

## 3 MAIN RESULTS

### 3.1 RESULT OF SDE-DPM-2

Before introducing the main results, we first give the following assumptions:

**Assumption 1.** *The data distribution has a bounded second moment, i.e., $\mathbb{E}_{p_0}\left[\|x\|^2\right] \leq M_2$.*

**Assumption 2.** *the estimated score function with Taylor expansion is $L^2$-accurate, i.e.,for all $k = 1, 2, \cdots, N$ and $t \in [t_{k-1}, t_k]$,*

$$\frac{1}{T}\sum_{k=1}^{N} h_k \mathbb{E}_{p_{t_k}} \| s\left(x_{t_k}, t_k\right) + \frac{\partial s(x_{t_k}, t_k)}{\partial t_k} \cdot (t - t_k) + J_{s_{t_k}} \cdot (x_t - x_{t_k})$$

$$- \nabla \log p_{t_k}(x_{t_k}) - \frac{\partial \nabla \log p_{t_k}(x_{t_k})}{\partial t_k} \cdot (t - t_k) - \nabla^2 \log p_{t_k}(x_{t_k}) \cdot (x_t - x_k) \|^2 \le \epsilon_0^2$$

**Assumption 3.** *The second-order derivative of the score function concerning $t$ are bounded, i.e., for all $k = 1, 2, \cdots, N$ and $t \in [t_{k-1}, t_k]$, there exist constants $C_1$ such that:*

$$\mathbb{E}_{p_t} \left\| \frac{\partial^2 \nabla \log p_t(x)}{\partial t^2} \right\|^2 \le C_1$$

*where $C$ is a constant independent of $t$ and only depends on the moments of the initial distribution $p_0$.*

**Assumption 4.** *The second-order derivative of the score function concerning $x$ are bounded, i.e., for all $k = 1, 2, \cdots, N$ and $t \in [t_{k-1}, t_k]$, there exists a constant $C_2$ such that:*

$$\mathbb{E}_{P_t} \left\| \nabla^3 \log p_t(x) \right\|^2 \le C_2$$

**Remark.** *The assumptions outlined in Assumption 1 are in line with that presented in Chen et al. (2023b;a). Additionally, the introduction of new assumptions, specifically Assumptions 2, 3, and 4, are designed for the analysis of second-order discretization methods.*

Before we present the main findings, let's delve into the reasoning that underpins Assumptions 2, 3, and 4.

Assumption 2 builds upon the $L^2$-accurate assumption from Chen et al. (2023b;a), which requires the estimated score function to exhibit $L^2$ accuracy:

$$\mathbb{E}_{p_{t_k}} \left[ \|s(x, t_k) - \nabla \log p_{t_k}(x)\|^2 \right] \le \epsilon_0^2$$

Assumption 2 presents a more stringent requirement. It demands that the Taylor expansion of the estimated score function exhibit $L^2$ accuracy. This heightened requirement is deemed justifiable, given the advancements in methodology proposed by Meng et al. (2021). Specifically, their work extends the utility of denoising score matching to the estimation of higher-order derivatives and empirically demonstrated that the first derivative of the score can be learned effectively under Gaussian mixture models. Such an approach significantly enhances the feasibility of accurately estimating the score function through its Taylor expansion.

To facilitate the Taylor expansion of the true score function, we introduce Assumptions 3 and 4. The core idea hinges on the premise that if the higher-order partial derivatives of the score function concerning $t$ and $x$ are bounded, then we can accurately estimate the Taylor expansion of the score function. In this paper, we demonstrate that Assumptions 3, and 4 hold under Gaussian Mixture distributions, which can approximate any smooth distributions and are widely used in practice. The constants $C_1$, $C_2$ only depend on the initial target distribution. See more details in Appendix E.

Now we introduce the main result of this paper. We give Theorem 3.1 to demonstrate the sampling complexity SDE-DPM-2 method:

**Theorem 3.1.** *under assumptions 1, 2, 3, and 4, SDE-DPM-2 has KL divergence bounded by:*

$$\text{KL}(p_0 || \hat{q}_T) \lesssim (M_2 + d)e^{-T} + T\epsilon_0^2 + \frac{C_2 d^3 T^3}{N^2}$$

*similarly, choosing $T = \log(\frac{M_2 + d}{\epsilon_0^2})$ and $N = \Theta(\frac{C_2^{0.5} d^{1.5} T^{1.5}}{\epsilon_0})$ makes the KL divergence $\tilde{O}(\epsilon_0^2)$.*

**Remark.** *The notation $\tilde{O}$ hides the logarithmic factors present in the sampling complexity. The gap between the estimated quantity $\hat{q}_k$ and the target distribution $p_0$ stems from three main sources: 1. The initial error, denoted as $(M_2 + d)e^{-T}$, originates from the starting point of the backward process, which assumes a normal distribution $\gamma_d = \mathcal{N}(0, I_d)$, instead of the desired distribution $p_T$. 2. The error associated with the score function estimation is expressed as $T\epsilon_0^2$. 3. The discretization*

method introduces an error quantified by $\frac{d^2 T^3}{N^2}$. There also exists a work Li et al. (2024) providing their results in terms of TV distance. By applying Pinsker's inequality: $TV(P,Q) \le \sqrt{KL(PQ)}$, our result $\tilde{O}(\epsilon_0^2)$ in KL divergence yields $\tilde{O}(\epsilon_0)$ in TV distance, and the sampling complexity to attain $\tilde{O}(\epsilon_0)$ in TV distance is $poly(d)/\epsilon$.

We will provide a detailed proof of Theorem 3.1 in the Appendix C.1. Note that we focus on KL divergence as the metric to bound the gap between $\hat{q}_T$ and $p_0$ like in several other works (Yang & Wibisono, 2022; Chen et al., 2023a; Benton et al., 2024). There is also work (Li et al., 2024) provide their result in terms of TV distance, and we will give a discussion in the Appendix A.2.

As a comparison, we also introduce Theorem 3.2 from Chen et al. (2023a) to demonstrate the sampling complexity of the EI method:

**Theorem 3.2** (Theorem 1 from Chen et al. (2023a)). *Assume that the target distribution $p_0$ and the estimated score function $s(x,t)$ satisfy*

1. $p_0$ *has a bounded second moment, i.e.,* $\mathbb{E}_{p_0}\left[\|x\|^2\right] \le M_2$.

2. $\nabla p_t(x_t)$ *is $L-$Lipschitz on $\mathbb{R}^d$.*

3. *the score function is $L^2$-accurate, i.e.,for all $k = 1, 2, \cdots, N$,*

$$\mathbb{E}_{p_{t_k}}\left[\|s(x, t_k) - \nabla \log p_{t_k}(x)\|^2\right] \le \epsilon_0^2$$

*then the KL divergence between the target distribution $p_0$ and the estimated distribution $\hat{q}_T$ generated by EI is bounded by:*

$$\text{KL}(p_0 || \hat{q}_T) \lesssim (M_2 + d)e^{-T} + T\epsilon_0^2 + \frac{d^2 T^2 L^2}{N} \tag{7}$$

*choosing $T = \log(\frac{M_2 + d}{\epsilon_0^2})$ and $N = \Theta(\frac{d^2 T^2 L^2}{\epsilon_0^2})$ makes the KL divergence $\tilde{O}(\epsilon_0^2)$.*

Comparing Theorem 3.1 and Theorem 3.2, the initial error and the estimation error of the score function are consistent with those in Theorem 3.2. However, we present the error associated with the discretization method as $\frac{C_2 d^3 T^3}{N^2}$, marking a significant improvement over the previously noted error of $\frac{d T^2 L^2}{N}$ for EI. This enhancement underscores the superior sampling complexity of SDE-DPM-2 compared to the EI method. Specifically, the sampling complexity of SDE-DPM-2 is $\tilde{O}(\frac{1}{\epsilon_0})$, which is notably more advantageous than the $\tilde{O}(\frac{1}{\epsilon_0^2})$ complexity of the EI method.

### 3.2 RESULT OF SECOND-ORDER RUNGE–KUTTA METHOD

Next, we will give the result of the second-order discretization method, the RK-2 method, and compare it with the SDE-DPM-2 method. RK-2 is another representative of second-order discretization methods, which is also referred to as the Heun's method for SDEs. To demonstrate RK-2, we rewrite the backward process (5) as follows:

$$\begin{aligned} \mathrm{d}x_t^{\leftarrow} &= (\hat{x}_t^{\leftarrow} + s(\hat{x}_k^{\leftarrow}, T-t))\,\mathrm{d}t + \sqrt{2}\mathrm{d}W_t \\ &= f(x_t^{\leftarrow}, t)\mathrm{d}t + \sqrt{2}\mathrm{d}W_t \end{aligned} \tag{8}$$

The update rule for the RK-2 method is given by:

$$\tilde{x}_{k+1}^{\leftarrow} = \hat{x}_k^{\leftarrow} + h_k f(\hat{x}_k^{\leftarrow}, t_k) \tag{9}$$

$$\hat{x}_{k+1}^{\leftarrow} = \hat{x}_k^{\leftarrow} + \frac{h_k}{2}\left(f(\hat{x}_k^{\leftarrow}, t_k) + f(\tilde{x}_{k+1}^{\leftarrow}, t_k + h_k)\right) + \sqrt{2h_k}z_k \tag{10}$$

where (9), (10) represent the predictor and corrector steps, respectively. In Lemma B.3, we demonstrate that the RK-2 method is equivalent to the following SDE (11):

$$\begin{aligned} \mathrm{d}x_t^{\leftarrow} &= \left(\hat{x}_{t_k}^{\leftarrow} + 2s(\hat{x}_k^{\leftarrow}, T-t_k)\right)\mathrm{d}t + \sqrt{2}\mathrm{d}W_t \\ &\quad + 2\left(\frac{\partial s(\hat{x}_{t_k}^{\leftarrow}, T-t_k)}{\partial t_k}(t-t_k) + J_{s_{t_k}}(x_t^{\leftarrow} - \hat{x}_{t_k}^{\leftarrow})\right)\mathrm{d}t \end{aligned} \tag{11}$$

Then we can give the following corollary:

**Corollary 3.3.** *Under Assumptions 1, 2, 3, and 4, the RK-2 method has KL divergence bounded by:*

$$\mathrm{KL}(p_0||\hat{q}_T) \lesssim (M_2 + d)e^{-T} + T\epsilon_0^2 + \frac{C_2 d^3 T^3}{N^2} + \frac{dT^2}{N}$$

*choosing $T = \log(\frac{M_2+d}{\epsilon_0^2})$ and $N = \Theta(\frac{C_2^{0.5} d^{1.5} T^{1.5}}{\epsilon_0} + \frac{dT^2}{\epsilon_0^2})$ makes the KL divergence $\tilde{O}(\epsilon_0^2)$.*

**Remark.** *A key distinction between RK-2 and SDE-DPM-2 is in their treatment of the linear component concerning $x_t$ of the drift term. While SDE-DPM-2 calculates the exact solution for this linear component, RK-2 opts for an approximation, starting from the initial value $\hat{x}_{t_k}$. This approximation strategy results in RK-2 exhibiting a KL divergence of $N = O(\frac{dT^2}{\epsilon_0^2})$ (only keeping the dominant order for $\epsilon$), which keeps the same order as EI.*

Theorems 3.2, 3.1, and Corollary 3.3 highlight the advanced sampling efficiency of the SDE-DPM-2 method when compared to the RK-2 and EI methods. SDE-DPM-2 demonstrates superior performance, achieving a sampling complexity of $\tilde{O}(\frac{1}{\epsilon_0})$, which is significantly more efficient than the $\tilde{O}(\frac{1}{\epsilon_0^2})$ complexity observed for both EI and RK-2. This enhanced efficiency is largely due to SDE-DPM-2's more precise approximation of the nonlinear component of the drift term, denoted as $s(x,t)$, over that of EI, and its exact solution for the linear component concerning $x_t$ of the drift term, in comparison to RK-2. Such improvements lead to a decrease in discretization error, markedly boosting sampling efficiency from complex distributions. To provide a comprehensive understanding of SDE-DPM-2, we also conduct comparisons of its properties with other SDE and ODE solvers in Appendix A.1.

## 4 PROOF SKETCH

Drawing inspiration from the work of Chen et al. (2023a), which explores the convergence properties of the EI method by decomposing the KL divergence between $p_0$ and $\hat{q}_T$ into three components, namely, the initial error, the score function error, and the discretization error, we intend to adopt a similar analytical framework.

Our focus will be on evaluating the convergence behavior of SDE-DPM-2. In Proposition 4.2, we identify and categorize the bound of KL divergence, $\mathrm{KL}(p_0, \hat{q}_T)$ into the initial error, the score function error, and the discretization error, when employing SDE-DPM-2. In Lemma 4.3, we characterize the discretization error of SDE-DPM-2. By combining these results, we can derive Theorem 3.1, which provides the sample complexity of SDE-DPM-2. The proof of Corollary 3.3 is similar to that of Theorem 3.1.

### 4.1 COMPARISON OF UPDATE SCHEMES

The EI scheme approximates the score function at time $[t_k, t_{k+1}]$ with $s_{T-t_k}(\hat{x}_k^{\leftarrow})$,

$$\mathrm{d}x_t^{\leftarrow} = (\hat{x}_t^{\leftarrow} + 2s(\hat{x}_k^{\leftarrow}, T - t_k))\,\mathrm{d}t + \sqrt{2}\mathrm{d}W_t \tag{12}$$

while the SDE-DPM-2 scheme approximates the score function at time $[t_k, t_{k+1}]$ with $s_{T-t_k}(\hat{x}_k^{\leftarrow}) + s^{(1)}(\hat{x}_k^{\leftarrow}, T - t_k)(t - t_k)$. Moreover, by denoting $s^{(1)}(\hat{x}_k^{\leftarrow}, T - t_k)(t - t_k)$ as:

$$s^{(1)}(\hat{x}_k^{\leftarrow}, T - t_k)(t - t_k) = \frac{\partial s(\hat{x}_{t_k}^{\leftarrow}, T - t_k)}{\partial t_k}(t - t_k) + J_{s_{t_k}} \cdot \frac{\partial \hat{x}_t^{\leftarrow}}{\partial t}(t - t_k)$$

$$= \frac{\partial s(\hat{x}_{t_k}^{\leftarrow}, t)}{\partial t_k}(t - t_k) + J_{s_{t_k}}(\hat{x}_t^{\leftarrow} - \hat{x}_{t_k}^{\leftarrow})$$

at each time interval $[t_k, t_{k+1}]$, the SDE-DPM-2 scheme can also be written as:

$$\begin{aligned}
\mathrm{d}x_t^{\leftarrow} = {} & (\hat{x}_t^{\leftarrow} + 2s(\hat{x}_k^{\leftarrow}, T - t_k))\,\mathrm{d}t + \sqrt{2}\mathrm{d}W_t \\
& + 2\left(\frac{\partial s(\hat{x}_{t_k}^{\leftarrow}, T - t_k)}{\partial t_k}(t - t_k) + J_{s_{t_k}}(x_t^{\leftarrow} - \hat{x}_{t_k}^{\leftarrow})\right)\mathrm{d}t
\end{aligned} \tag{13}$$

## 4.2 KL Divergence Decomposition

With (12), Chen et al. (2023a) proposed that the KL divergence between $p_0$ and $\hat{q}_T$ when employing the EI method is constrained by the initial error, the score estimation error, and the error due to discretization, as detailed in Proposition 4.1.

**Proposition 4.1** (proposition 8 from Chen et al. (2023a)). *if the estimated score function $s(x, t)$ satisfies*

$$\mathbb{E}_{p_{t_k}}\left[\|s(x, t_k) - \nabla \log p_{t_k}(x)\|^2\right] \leq \epsilon_0^2$$

*the KL divergence between $p_0$ and $\hat{q}_T$ with EI method is bounded by:*

$$\mathrm{KL}\left(p_0\|\hat{q}_T\right) \leq \mathrm{KL}\left(p_T\|\gamma_d\right) + T\epsilon_0^2 + \sum_{k=1}^{N}\int_{t_{k-1}}^{t_k}\mathbb{E}\left\|\nabla \log p_{t_k}\left(x_{t_k}\right) - \nabla \log p_t\left(x_t\right)\right\|^2 \mathrm{d}t$$

Proposition 4.1 bound the KL divergence between $p_0$ and $\hat{q}_T$ with EI method by the initial error: $\mathrm{KL}\left(p_T\|\gamma_d\right)$, the error of the score function: $T\epsilon_0^2$, and the discretization error: $\sum_{k=1}^{N}\int_{t_{k-1}}^{t_k}\mathbb{E}\left\|\nabla \log p_{t_k}\left(x_{t_k}\right) - \nabla \log p_t\left(x_t\right)\right\|^2 \mathrm{d}t$.

Building on this framework, by employing (13), the KL divergence between $p_0$ and $\hat{q}_T$ when using the DPM-2 method is similarly limited by the initial error, the score function error, and the discretization error, which is elaborated in Proposition 4.2 as follows:

**Proposition 4.2.** *With Assumption 2, the KL divergence between $p_0$ and $\hat{q}_T$ with DPM-2 method is bounded by:*

$$\mathrm{KL}\left(p_0\|\hat{q}_T\right) \leq \mathrm{KL}\left(p_T\|\gamma_d\right) + T\epsilon_0^2$$

$$+ \sum_{k=1}^{N}\int_{t_{k-1}}^{t_k}\mathbb{E}\|\nabla \log p_{t_k}(x_{t_k}) + \frac{\partial \nabla \log p_{t_k}(x_{t_k})}{\partial t_k}(t - t_k)$$

$$+ \nabla^2 \log p_{t_k}(x_{t_k})(x_t - x_k) - \nabla \log p_t(x_t)\|^2 \mathrm{d}t$$

See the derivation of Proposition 4.2 in Appendix C.1.

When examining Proposition 4.1 alongside Proposition 4.2, a key difference between the EI and DPM-2 methodologies becomes evident, particularly in the context of their discretization errors. Specifically, the discretization error associated with the EI method is characterized as follows:

$$\int_{t_{k-1}}^{t_k}\mathbb{E}\left\|\nabla \log p_{t_k}\left(x_{t_k}\right) - \nabla \log p_t\left(x_t\right)\right\|^2 \mathrm{d}t \tag{14}$$

whereas for the DPM-2 method, it is characterized as:

$$\int_{t_{k-1}}^{t_k}\mathbb{E}\|\nabla \log p_{t_k}(x_{t_k}) + \frac{\partial \nabla \log p_{t_k}(x_{t_k})}{\partial t_k}\cdot(t - t_k) + \nabla_{x_{t_k}}^2 \log p_{t_k}(x_{t_k})\cdot(x_t - x_k) - \nabla \log p_t(x_t)\|^2 \mathrm{d}t \tag{15}$$

To derive Theorem 3.1, it is necessary to establish the discretization error for DPM-2. To this end, we introduce the following lemma to analyze the discretization error of DPM-2. Specifically, the discretization error of DPM-2 is bounded by Lemma 4.3:

**Lemma 4.3.** *under Assumptions 1, 3, 4, the discretization error of DPM-2 ((15)) is bounded by:*

$$\int_{t_{k-1}}^{t_k}\mathbb{E}\|\nabla \log p_{t_k}(x_{t_k}) + \frac{\partial \nabla \log p_{t_k}(x_{t_k})}{\partial t_k}\cdot(t - t_k)$$

$$+ \nabla_{x_{t_k}}^2 \log p_{t_k}(x_{t_k})\cdot(x_t - x_k) - \nabla \log p_t(x_t)\|^2 \mathrm{d}t \tag{16}$$

$$\lesssim C_2 d^3 h_k^3$$

The proof of the lemma is detailed in Appendix C.1. And the initial error term $\mathrm{KL}\left(p_T\|\gamma_d\right)$ is bounded in Lemma B.2.

By substituting (16) into Proposition 4.2, we can derive the main theorem, Theorem 3.1. As for Corollary 3.3, Comparing 13 and 11, RK-2 will induce an additional term $\mathbb{E}\left\|x_t - x_{t_k}\right\|^2$, which will lead to a higher discretization error. See the detailed proof in Appendix C.2.

## 5 DISCUSSION OF VE-SDE DPMs

We will briefly discuss that our analysis can be extended to the VE-SDE DPMs. The SE-SDE forward process is as follows,

$$\mathrm{d}x = \sqrt{\frac{\partial \sigma^2(t)}{\partial t}}\mathrm{d}w \tag{17}$$

following (17), the conditional distribution of $x_t$ given $x_0$ is $p_t(x_t|x_0) \sim \mathcal{N}(x_0, \sigma^2(t)I_d)$.

We hereafter adopt the settings in Chen et al. (2023c), where $\sigma^2(t) = 2t$, the forward process can be written as:

$$\mathrm{d}x = \sqrt{2}\mathrm{d}w$$

then the backward process can be written as:

$$\mathrm{d}x_t^{\leftarrow} = 2 \cdot s(x_t^{\leftarrow}, T - t)\mathrm{d}t + \sqrt{2}\mathrm{d}w_t \tag{18}$$

When comparing (18) with (5), the notable distinction arises in the drift term, specifically in the $x_t^{\leftarrow}$ component. Consequently, the approach to deriving the discretization error for VE-SDE DPMs closely aligns with that employed for VP-SDE DPMs. It is observed that the discretization error for both SDE-DPM-2 and RK-2 under the VE-SDE framework mirrors the previously established discretization error term found in (15). From this observation, we can draw the following corollary:

**Corollary 5.1.** *Under Assumptions 1 2, and if Assumptions 3, 4 also hold for VE-SDE (17), the KL divergence between $p_0$ and $\hat{q}_T$ with DPM-2 or RK-2 method is bounded by:*

$$\mathrm{KL}\left(p_0 \| \hat{q}_T\right) \leq \mathrm{KL}\left(p_T \| \gamma_d\right) + T\epsilon_0^2 + \frac{C_2 d^3 T^3}{N^2}$$

**Remark.** *In the context of equation (18), the KL divergence $\mathrm{KL}(p_T|\gamma_d)$ converges at a rate of only $\frac{1}{T}$, as shown by Lee et al. (2022). This rate is significantly slower than the exponential $e^{-T}$ convergence observed under equation (1). Consequently, it may dominate the discretization error term of $1/T^2$ for both SDE-DPM-2 and RK-2. If we disregard the initial error by assuming that the backward process starts directly at $p_T$, then setting $T = \log(\frac{1}{\epsilon_0^2})$ and $N = \Theta\left(\frac{C_2^{0.5}d^{1.5}T^{1.5}}{\epsilon_0}\right)$ reduces the KL divergence to $\tilde{O}(\epsilon_0^2)$. A detailed proof can be found in Appendix D.*

Corollary 5.1 reveals that when utilizing the SDE-DPM2 method, VE-SDE DPMs achieve a convergence order comparable to that of VP-SDE DPMs. This result highlights the consistency and effectiveness of the SDE-DPM-2 approach across various diffusion models frameworks.

## 6 EMPIRICAL RESULTS

In order to present the practical scaling of our main result theorems more clearly and intuitively, we conduct experiments on Gaussian mixture and CIFAR-10 dataset. Fig. 1a empirically shows the KL divergence of SDE-DPM-2 and other methods under different discretization numbers. SDE-DPM-2 demonstrates a faster decrease rate compared to other solvers. Fig. 1b shows that the empirical results and the theoretical results in the logarithmic scale. It is observed that RK-2 and SDE-DPM show comparable empirical performance, both are less efficient than SDE-DPM-2. Moreover, we identified a gap between existing theoretical results and empirical observations, as the KL divergence for each method decreases more rapidly than the theoretical bounds. This may suggest potential directions for future research to further improve the convergence rate of these methods.

While Lu et al. (2022a) demonstrated that SDE-DPM-2 generates better samples than SDE-DPM through the image generation examples, we directly compare the FID score of SDE-DPM-2 and SDE-DPM to further validate the effectiveness of the second-order method. We implement both solvers to sample images from a pretrained model based on DDPM on CIFAR-10. Table 1 provides additional support, which further substantiates the improved experimental performance of SDE-DPM-2 over SDE-DPM, offering a more detailed empirical validation of its effectiveness.

Considering computational cost, SDE-DPM-2 efficiently updates the derivative of the score function by leveraging previously stored results. This optimization ensures that the number of score function

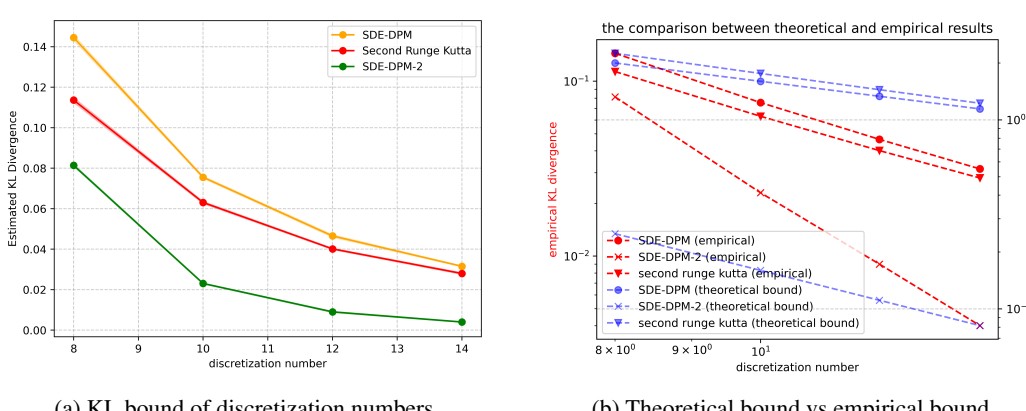

(a) KL bound of discretization numbers      (b) Theoretical bound vs empirical bound

Figure 1: Results on Gaussian mixture

| Sampling Steps | SDE-DPM-2 | SDE-DPM |
|---|---|---|
| 20 | $17.98 \pm 0.023$ | $33.24 \pm 0.097$ |
| 30 | $15.19 \pm 0.117$ | $25.20 \pm 0.287$ |
| 50 | $13.66 \pm 0.110$ | $19.75 \pm 0.166$ |
| 100 | $12.96 \pm 0.089$ | $15.48 \pm 0.155$ |

Table 1: FID score on CIFAR-10 with different sampling steps. Each result is averaged over 5 runs.

evaluations remains the same as in SDE-DPM. For example, in the CIFAR-10 experiment, sampling 20,000 images takes approximately 753 seconds with SDE-DPM and 765 seconds with SDE-DPM-2. The additional computational cost of SDE-DPM-2 is negligible.

# 7 CONCLUSION AND DISCUSSION

We conduct a detailed examination of $\mathrm{KL}(p_0, \hat{q}_T)$, focusing on the SDE-DPM-2 method. Our analysis reveals that the SDE-DPM-2 method significantly outperforms the EI method in terms of sampling complexity. Specifically, the sampling complexity for the SDE-DPM-2 method is $\tilde{O}(\frac{1}{\epsilon_0})$, which is more efficient compared to the EI method's $\tilde{O}(\frac{1}{\epsilon_0^2})$. Additionally, we also analyze the RK-2 method, which involves a direct discretization of the linear component concerning $x_t$ of the drift term in the reverse SDE. We find that it necessitates a sampling complexity of $\tilde{O}(\frac{1}{\epsilon_0^2})$. This indicates a lower efficiency than the SDE-DPM-2 method, primarily due to the increased discretization error associated with the RK-2 method's direct approach to discretizing the linear drift term. Our findings underscore the superior efficiency of the SDE-DPM-2 method over both the EI and RK-2 methods in terms of sampling complexity.

Furthermore, we delved into the convergence behavior of the SDE-DPM-2 method within the VE-SDE framework, finding that its convergence characteristics are consistent with those observed in the VP-SDE framework. This consistency highlights the SDE-DPM-2 method's adaptability across different diffusion models frameworks.

In this study, our focus was solely on the second-order discretization method, namely SDE-DPM-2. Future studies could investigate the convergence properties of higher-order discretization methods, such as SDE-DPM-3, to see how they compare in efficiency with the SDE-DPM-2 method. Our discussion was limited to Assumptions 3 and 4, considering the context of Gaussian distributions and Gaussian Mixture Models (GMMs) within the Variance Preserving (VP) SDE framework. We leave the examination of Assumptions 3 and 4 in more general scenarios, including non-Gaussian distributions for future research.

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

# A    COMPARISON WITH OTHER SOLVERS

## A.1    COMPARISON OF SDE SOLVER PROPERTIES

We first introduce two key convergence measures for the approximation performance of SDE solvers: Strong Order and Weak Order.

**Definition A.1** (Strong Order, Definition 1.1 from Wang (2016))**.** *Suppose $y$ is the discrete-time approximation of the solution $x(t)$ of the SDE, and $h$ is the step size. The strong order of the method is $p$ if there exists a constant $C$ such that*

$$\mathbb{E}\left[\|x(T) - y(T)\|\right] \leq Ch^p$$

**Definition A.2** (Weak Order, Definition 1.2 from Wang (2016))**.** *Suppose $y$ is the discrete-time approximation of the solution $x(t)$ of the SDE, and $h$ is the step size. The weak order of the method is $p$ if there exists a constant $C$ such that*

$$\|\mathbb{E}\left[x(T)\right] - \mathbb{E}\left[y(T)\right]\| \leq Ch^p$$

To comprehensively analyze the properties of the SDE-DPM-2 solver, Table 2 compares weak order, strong order, and KL divergence order across various SDE solvers. Notably, some solvers, such as SRA3 from Rößler (2010), exhibit higher weak and strong orders than SDE-DPM-2. However, the sampling complexity of these methods remains unexplored. Investigating the sampling complexity of such methods could be an intriguing direction for future work.

| Solvers | Weak Order | Strong Order | KL Order |
|---|---|---|---|
| EM | 1 | 0.5 | 1 |
| SDE-DPM | 1 | 0.5 | 1 |
| SDE-DPM-2 | 2 | 1 | 2 |
| Heun | 2 | 1 | 1 |
| SRA1/2 from Rößler (2010) | 2 | 1.5 | - |
| SRA3 from Rößler (2010) | 3 | 1.5 | - |
| Stochastic Ralston method from Foster et al. (2024) | 2 | 1.5 | - |

Table 2: Comparison of solvers with their respective weak, strong, and KL orders.

As detailed in Proposition 4.2 of the manuscript, the KL divergence could be decomposed into the sum of the initial error, the estimation error of the score function, and the discretization error of the drift terms of the Reverse SDE and the SDE induced by SDE-solvers. The forward process's convergence property (Lemma B.2) and the assumption of the $L^2$ accuracy of the score function help control the initial error and the estimation error of the score function. The main focus is on the discretization error of the drift term. The drift term of the reverse SDE is $\nabla \log p_{t_k}(x_{t_k}) + x_t$, and in the manuscript, the drift terms of the SDE solvers are listed as follows:

| solvers | drift term |
|---|---|
| EM | $\nabla \log p_{t_k}(x_{t_k}) + x_{t_k}$ |
| SDE-DPM | $\nabla \log p_{t_k}(x_{t_k}) + x_t$ |
| SDE-DPM-2 | $\nabla \log p_{t_k}(x_{t_k}) + \frac{\partial \nabla \log p_{t_k}(x_{t_k})}{\partial t_k}(t - t_k) + x_t$ |
| RK-2 (Heun) | $\nabla \log p_{t_k}(x_{t_k}) + \frac{\partial \nabla \log p_{t_k}(x_{t_k})}{\partial t_k}(t - t_k) + x_{t_k}$ |

Table 3: Comparison of solvers and their corresponding drift terms

Both SDE-DPM-Solver and DPM-Solver-2 utilize the linearity of the drift term of the reverse SDE, which eliminates the error of $x_t - x_{t_k}$.

In comparison to solvers with weak order 1, the solvers with weak order 2 introduce an additional term $\frac{\partial \nabla \log p_{t_k}(x_{t_k})}{\partial t_k}(t - t_k)$, which reduces the discretization error of the score function.

It can be inferred that SDE solvers with higher weak/strong order, such as SRA1 from Rößler (2010) and the stochastic Ralston method from Foster et al. (2024), might introduce higher-order terms of derivatives of the score function. These could potentially reduce the discretization error of the score function. However, these methods might still retain the dominating term $x_t - x_{t_k}$ in the drift term's discretization error. Consequently, due to the current analytical approaches prevalent in the diffusion models community, it appears that we are limited to achieving a KL divergence order of 1 for higher-order SDE solvers, such as SRA1 from Rößler (2010). Exploring the potential of higher-order SDE solvers with alternative analytical approaches could be an interesting direction for future work.

It is important to consider the potential increase in computational cost associated with higher-order solvers, particularly due to the increased number of score function evaluations, which is the most computationally expensive part of the diffusion models sampling process. Higher-order solvers may lead to an increase in the number of score function evaluations, thereby escalating the computational cost. This could be a factor that might require further investigation to balance convergence rate and computational efficiency. Lu et al. (2022b) designed the SDE-DPM-Solver++(2M), which efficiently updates the derivative of the score function using previously stored results, thereby maintaining the computational cost equivalent to that of a first-order method.

We also provide the comparison of sampling complexity of both SDE and ODE solvers in table 4.

| Solvers | Sampling Complexity to attain $\tilde{O}(\epsilon^2)$ in KL divergence |
|---|---|
| ODE: DDIM/ODE-EI | $poly(d)/\epsilon$ (Li et al., 2024) |
| ODE-DPM-2 | $poly(d)/\sqrt{\epsilon}$ (Li et al., 2024) |
| SDE-EI | $poly(d)/\epsilon^2$ (Chen et al., 2023a) |
| SDE-DPM-2 | $poly(d)/\epsilon$ (Ours) |

Table 4: Comparison of sampling complexity of solvers.

## A.2 DISCUSSION OF KL DIVERGENCE AND TV DISTANCE BOUND

As we mention in Section 3.1, our $\tilde{O}(\epsilon^2)$ result in KL divergence yields $\tilde{O}(\epsilon)$ in TV distance. Therefore, our result for the sampling complexity of SDE-DPM-2 solver is $\tilde{O}(\frac{1}{\epsilon})$ to attain $\tilde{O}(\epsilon)$ TV distance. Table 5 compares the KL divergence and TV distance of the solvers:

| solvers | sample complexity attaining $\tilde{O}(\epsilon)$ in KL divergence | sample complexity attaining $\tilde{O}(\epsilon)$ in TV distance |
|---|---|---|
| SDE-DPM-Solver | $poly(d)/\epsilon$ (Yang & Wibisono, 2022) | $poly(d)/\epsilon^2$ (Chen et al., 2023b) |
| SDE-DPM-Solver-2 | $poly(d)/\sqrt{\epsilon}$ (Ours) | $poly(d)/\epsilon$ (Ours) |
| second-order SDE solver in Li et al. (2024) | - | $poly(d)/\epsilon$ (Li et al., 2024) |

Table 5: KL divergence and TV distance of the solvers.

## B  USEFUL LEMMAS

Unless specifically noted otherwise, the lemmas discussed are developed within the framework of the VP-SDE. Lemma B.1 establishes bounds on the expected values of $\|x_t - x_{t_k}\|^2$ and $\|x_t\|^2$. Furthermore, Lemma B.2 sets a limit on the Kullback-Leibler divergence between $p_T$ and $\gamma_d$. Finally, Lemma B.3 demonstrates that the Runge-Kutta-2 update scheme is equivalent to a specific SDE.

**Lemma B.1** (lemma 10 from Chen et al. (2023b)). *Under Assumption 1, suppose that $h_k \leq 1$ for $1 \leq k \leq N$, for $t_{k-1} \leq t \leq t_k$, we have*

$$\mathbb{E}\|x_t - x_{t_k}\|^2 \lesssim d(t_k - t) + (M_2 + d)(t_k - t)^2,$$

*and*

$$\mathbb{E}\|x_t\|^2 \leq d + M_2$$

**Lemma B.2** (lemma 9 from Chen et al. (2023a) ). *Under Assumption 1, the initial error is bounded by:*

$$\text{KL}(p_T||\gamma_d) \leq (M_2 + d)e^{-T}$$

**Lemma B.3.** *the update scheme of Runge-Kutta-2 at each time interval $[t_k, t_{k+1}]$, is equalvent to the following SDE:*

$$\begin{aligned}
\mathrm{d}x_t^{\leftarrow} = {}& \left(\hat{x}_{t_k}^{\leftarrow} + 2s(\hat{x}_k^{\leftarrow}, T - t_k)\right)\mathrm{d}t + \sqrt{2}\mathrm{d}W_t \\
& + 2\left(\frac{\partial s(\hat{x}_{t_k}^{\leftarrow}, T - t_k)}{\partial t_k}(t - t_k) + \frac{\partial s(\hat{x}_{t_k}^{\leftarrow}, T - t)}{\partial \hat{x}_{t_k}^{\leftarrow}}(x_t^{\leftarrow} - \hat{x}_{t_k}^{\leftarrow})\right)\mathrm{d}t
\end{aligned} \tag{19}$$

*proof of Lemma B.3.* Given the following SDE:

$$\mathrm{d}x_t^{\leftarrow} = f(x_t^{\leftarrow}, t)\mathrm{d}t + \sqrt{2}\mathrm{d}W_t$$

we approximate $f(\tilde{x}_{k+1}^{\leftarrow}, t_k + h_k)$ as:

$$f(\tilde{x}_{k+1}^{\leftarrow}, t_k + h_k) = f(\hat{x}_k^{\leftarrow}, t_k) + h_k\frac{f}{\partial t_k} + h_k f(\hat{x}_k^{\leftarrow}, t_k)\frac{\partial f}{\partial \hat{x}_{t_k}^{\leftarrow}} + O(h^2)$$

, then substituting this into the corrector step (10), we have:

$$\hat{x}_{k+1}^{\leftarrow} = \hat{x}_k^{\leftarrow} + \frac{h_k}{2}\left(2f(\hat{x}_k^{\leftarrow}, t_k) + h_k\frac{f}{\partial t_k} + h_k f(\hat{x}_k^{\leftarrow}, t_k)\frac{\partial f}{\partial \hat{x}_{t_k}^{\leftarrow}}\right) + \sqrt{2h_k}z_k + O(h_k^3)$$

simplifying, we have:

$$\hat{x}_{t_k+h_k}^{\leftarrow} = \hat{x}_{t_k}^{\leftarrow} + h_k f(x_{t_k}^{\leftarrow}, t_k) + \frac{h_k^2}{2}(\frac{\partial f}{\partial t_k} + f(x_{t_k}^{\leftarrow}, t_k)\frac{\partial f}{\partial \hat{x}_{t_k}^{\leftarrow}}) + \sqrt{2h_k}z_k + O(h^3) \tag{20}$$

With (20), and $f(x_{t_k}^{\leftarrow}, t_k) = x_{t_k}^{\leftarrow} + 2s(x_{t_k}^{\leftarrow}, T - t_k)$, we complete the proof. $\qquad\square$

**Lemma B.4.** *for $\forall t_k, t \in [0, T]$, we have the following inequality:*

$$\mathbb{E}_{p_t}\|x_t\|^4 \leq M_4 + d^2 + d$$

$$\mathbb{E}_{p_t}\|x_t - x_k\|^4 \leq d^2(t - t_k)^2 + (M_4 + d^2 + d)(t - t_k)^2$$

*where $M_4 := \mathbb{E}_{p_0}\|x_0\|^4$.*

*Proof.* since we have $x_t = e^{-t}x_0 + \sqrt{1 - e^{-2t}}z$, where $z \sim \mathcal{N}(0, I_d)$, we have

$$\begin{aligned}
\mathbb{E}_{p_t}\|x_t\|^4 &= \mathbb{E}_{p_0}\left\|e^{-t}x_0 + \sqrt{1 - e^{-2t}}z\right\|^4 \\
&\leq 3\mathbb{E}_{p_0}\left\|e^{-t}x_0\right\|^4 + 3\mathbb{E}_{p_0}\left\|\sqrt{1 - e^{-2t}}z\right\|^4 \\
&\leq 3e^{-4t}\mathbb{E}_{p_0}\|x_0\|^4 + 3(1 - e^{-2t})^2(d^2 + d) \\
&\leq M_4 + d^2 + d
\end{aligned}$$

and for the second inequality, we have

$$\begin{aligned}
\mathbb{E}_{p_t}\|x_t - x_k\|^4 &= \mathbb{E}\left\|\int_t^{t_k} x_u \mathrm{d}u + \int_t^{t_k'} \sqrt{2}\mathrm{d}w_u\right\|^4 \\
&\lesssim \mathbb{E}\left\|\int_t^{t_k} x_u \mathrm{d}u\right\|^4 + \mathbb{E}\left\|\int_t^{t_k'} \sqrt{2}\mathrm{d}w_u\right\|^4 \\
&\leq (t_k - t)\left(\int_t^{t_k} \mathbb{E}\|x_u\|^4 \mathrm{d}u\right) + d^2(t - t_k)^2 \\
&\leq (t_k - t)^2(M_4 + d^2 + d) + d^2(t - t_k)^2
\end{aligned}$$

$$\square$$

## C    PROOFS FOR THE MAIN THEOREMS

### C.1    PROOF OF THEOREM 3.1

We will first give the proof of Proposition 4.2, and then provide the proof of Lemma 4.3.

PROOF OF PROPOSITION 4.2

The proof of Proposition 4.2 is similar to that of Proposition 4.1 as presented by Chen et al. (2023a). By replacing the EI update scheme (12) with the SDE-DPM-2 scheme (13), we render the proof of Proposition 4.2 straightforward.

*Proof.* For $0 \leq j \leq N - 1$, let $t'_k = T - t_k$, considering the following SDEs starting from $x_{t'_k} = a$, for time $t \in (t'_k, t'_{k+1}]$:

$$\mathrm{d}x_t = [x_t + 2\nabla \log p_t(x_t)]\, \mathrm{d}t + \sqrt{2}\mathrm{d}W_t, \quad x_{t'_k} = a \tag{21}$$

and its corresponding discretization approximation with $\nabla \log p_t(x_t) = s(a, t'_k) + s^{(1)}(a, t'_k) \cdot (t - t'_k)$:

$$\mathrm{d}y_t = \left[ y_{t_k} + 2s(a, T - t_k) + 2s^{(1)}(a, T - t_k) \cdot (t - t'_k) \right] \mathrm{d}t + \sqrt{2}\mathrm{d}W_t, \quad y_{t'_k} = a \tag{22}$$

Let $x_t, y_t$ admit densities $p_t, q_t$, respectively. With Proposition 8 in Chen et al. (2023a), we have

$$\frac{\mathrm{d}}{\mathrm{d}t} \mathrm{KL}\left( p_{t|t'_k}(\cdot \mid a) \| \hat{q}_{t|t'_k}(\cdot \mid a) \right)$$

$$= -2\mathbb{E}_{p_{t|t'_k}(y|a)} \left\| \nabla \log \frac{p_{t|t'_k}(y \mid a)}{q_{t|t'_k}(y \mid a)} \right\|^2$$

$$+ \mathbb{E}_{p_{t|t'_k}(y|a)} \left[ \left\langle \left( 2\nabla \log p_t(y) - 2s\left( a, t_{N-k} \right) - 2s^{(1)}(a, T - t_k) \cdot (t - t'_k) \right), \nabla \log \frac{p_{t|t'_k}(y \mid a)}{q_{t|t'_k}(y \mid a)} \right\rangle \right]$$

$$\leq \mathbb{E}_{p_{t|t'_k}(y|a)} \left\| s(a, T - t_k) + s^{(1)}(a, T - t_k) \cdot (t - t'_k) - \nabla \log p_t(y) \right\|^2,$$

where the last inequality follows from the Cauchy-Schwarz inequality. Integrating the above inequality from $t'_k$ to $t'_{k+1}$, we have

$$\mathrm{KL}(p_{t'_{k+1}}(\cdot|a) \| \hat{q}_{t'_{k+1}}(\cdot|a)) \leq \int_{t'_k}^{t'_{k+1}} \mathbb{E}_{p_{t|t'_k}(y|a)} \left\| s(a, T - t_k) + s^{(1)}(a, T - t_k) \cdot (t - t'_k) - \nabla \log p_t(y) \right\|^2 \mathrm{d}t.$$

Then for each $k \in [0, 1, \cdots, N - 1]$, using chain rule in of KL divergence, we have

$$\mathrm{KL}(p_{t'_{k+1}} \| q_{t'_{k+1}}) \leq \mathbb{E}_{p_{t'_k}(a)} \mathrm{KL}\left( p'_{t'_{k+1}} \Big| t'_k(\cdot \mid a) \| q'_{t'_{k+1}} \Big| t'_k(\cdot \mid a) \right) + \mathrm{KL}\left( p_{t'_k} \| q_{t'_k} \right)$$

$$\leq \mathrm{KL}\left( p_{t'_k} \| q_{t'_k} \right)$$

$$+ \int_{t'_k}^{t'_{k+1}} \mathbb{E}_{p_t(y)} \left\| s(a, T - t_k) + s^{(1)}(a, T - t_k) \cdot (t - t'_k) - \nabla \log p_t(y) \right\|^2 \mathrm{d}t.$$

summation over $k$ yields

$$
\begin{aligned}
\mathrm{KL}\left(p_0 \| \hat{q}_T\right) \leq{} & \mathrm{KL}\left(p_T \| \gamma_d\right)+\sum_{k=0}^{N-1} \int_{t_k'}^{t_{k+1}'} \mathbb{E}_{p_t(y)}\left\|s(a, T-t_k)+s^{(1)}(a, T-t_k) \cdot(t-t_k')-\nabla \log p_t(y)\right\|^2 \mathrm{~d}t \\
={} & \mathrm{KL}\left(p_T \| \gamma_d\right)+\sum_{k=1}^{N} \mathbb{E}_{p_{t_k}}\left\|s\left(x_{t_k}, t_k\right)+s^{(1)} \cdot(t-t_k)-\nabla \log p_t(x_t)\right\|^2 \mathrm{~d}t \\
={} & \mathrm{KL}\left(p_T \| \gamma_d\right)+\sum_{k=1}^{N} \mathbb{E}_{p_{t_k}}\left\|s\left(x_{t_k}, t_k\right)+\frac{\partial s(x_{t_k}, t_k)}{\partial t_k}(t-t_k)+J_{s_{t_k}}(x_t-x_{t_k})-\nabla \log p_t(x_t)\right\|^2 \mathrm{~d}t \\
\leq{} & \mathrm{KL}\left(p_T \| \gamma_d\right) \\
& +\sum_{k=1}^{N} \mathbb{E}\left\|s\left(x_{t_k}, t_k\right)+\frac{\partial s(x_{t_k}, t_k)}{\partial t_k}(t-t_k)+J_{s_{t_k}}(x_t-x_{t_k})\right. \\
& \left.-\nabla \log p_{t_k}(x_{t_k})-\frac{\partial \nabla \log p_{t_k}(x_{t_k})}{\partial t_k}(t-t_k)-\nabla^2 \log p_{t_k}(x_{t_k})(x_t-x_k)\right\|^2 \mathrm{~d}t \\
& +\sum_{k=1}^{N} \int_{t_{k-1}}^{t_k} \mathbb{E}\left\|\nabla \log p_{t_k}(x_{t_k})+\frac{\partial \nabla \log p_{t_k}(x_{t_k})}{\partial t_k}(t-t_k)\right. \\
& \left.+\nabla^2 \log p_{t_k}(x_{t_k})(x_t-x_k)-\nabla \log p_t(x_t)\right\|^2 \mathrm{~d}t \\
\leq{} & \mathrm{KL}\left(p_T \| \gamma_d\right)+T \epsilon_0^2 \\
& +\sum_{k=1}^{N} \int_{t_{k-1}}^{t_k} \mathbb{E}\left\|\nabla \log p_{t_k}(x_{t_k})+\frac{\partial \nabla \log p_{t_k}(x_{t_k})}{\partial t_k}(t-t_k)\right. \\
& \left.+\nabla^2 \log p_{t_k}(x_{t_k})(x_t-x_k)-\nabla \log p_t(x_t)\right\|^2 \mathrm{~d}t
\end{aligned}
$$

$\square$

PROOF OF LEMMA 4.3

*Proof.* With Taylor's Formula, the score function concering $t$ for an given $x_{t_k}$ can be approximated as:

$$
\nabla \log p_t(x_{t_k})=\nabla \log p_{t_k}(x_{t_k})+\frac{\partial \nabla \log p_{t_k}(x_{t_k})}{\partial t_k}(t-t_k)+\frac{\partial^2 \nabla \log p_s(x_{t_k})}{\partial s^2}(t-t_k)^2 \quad (23)
$$

Similarly, the score function concering $x$ for an given $t$ can be approximated as:

$$
\nabla \log p_t(x)=\nabla \log p_t(x_{t_k})+\nabla_{x_{t_k}}^2 \log p_t(x_{t_k}) \cdot(x-x_{t_k})+I_d \otimes(x-x_{t_k}) \nabla_{x_{t_s}}^3 \log p_t(x_{t_s}) \cdot(x-x_{t_k}) \quad (24)
$$

where $s \in[t_k, t]$.

We divide (15) into two parts:

$$
\begin{aligned}
\int_{t_{k-1}}^{t_k} \mathbb{E}&\left\|\nabla \log p_{t_k}(x_{t_k})+\frac{\partial \nabla \log p_{t_k}(x_{t_k})}{\partial t_k} \cdot(t-t_k)\right. \\
& \left.+\nabla_{x_{t_k}}^2 \log p_{t_k}(x_{t_k}) \cdot(x_t-x_k)-\nabla \log p_t(x_t)\right\|^2 \mathrm{~d}t \\
\leq \int_{t_{k-1}}^{t_k} \mathbb{E}&\left\|\nabla \log p_{t_k}(x_{t_k})+\frac{\partial \nabla \log p_{t_k}(x_{t_k})}{\partial t_k} \cdot(t-t_k)-\nabla \log p_t(x_{t_k})\right\|^2 \mathrm{~d}t \quad (25) \\
+\int_{t_{k-1}}^{t_k} \mathbb{E}&\left\|\nabla \log p_t(x_{t_k})+\nabla_{x_{t_k}}^2 \log p_t(x_{t_k}) \cdot(x_t-x_k)-\nabla \log p_t(x_t)\right\|^2 \mathrm{~d}t \quad (26)
\end{aligned}
$$

where (25) and (26) are due to the triangle inequality. For (25), we have

$$\int_{t_{k-1}}^{t_k} \mathbb{E} \left\| \nabla \log p_{t_k}(x_{t_k}) + \frac{\partial \nabla \log p_{t_k}(x_{t_k})}{\partial t_k} \cdot (t - t_k) - \nabla \log p_t(x_{t_k}) \right\|^2 \mathrm{d}t$$

$$\leq \int_{t_{k-1}}^{t_k} \mathbb{E} \left\| \frac{\partial^2 \nabla \log p_s(x_{t_k})}{\partial s^2}(t - t_k)^2 \right\|^2 \mathrm{d}t \tag{27}$$

$$\leq \int_{t_{k-1}}^{t_k} C_1(t - t_k)^4 \mathrm{d}t \tag{28}$$

$$\lesssim C_1 h_k^5 \tag{29}$$

- (27) is derived from (23),

- (28) is derived from Lemma E.1.

For (26), we have

$$\int_{t_{k-1}}^{t_k} \mathbb{E} \left\| \nabla \log p_t(x_{t_k}) + \nabla^2_{x_{t_k}} \log p_t(x_{t_k}) \cdot (x_t - x_k) - \nabla \log p_t(x_t) \right\|^2 \mathrm{d}t$$

$$\leq \int_{t_{k-1}}^{t_k} \mathbb{E} \left\| I_d \otimes (x_t - x_{t_k}) \nabla^3_{x_{t_s}} \log p_t(x_{t_s}) \cdot (x_t - x_{t_k}) \right\|^2 \mathrm{d}t \tag{30}$$

$$\leq \int_{t_{k-1}}^{t_k} d \cdot C_2 \mathbb{E} \left\| x_t - x_{t_k} \right\|^4 \mathrm{d}t \tag{31}$$

$$\lesssim d \cdot C_2 \left( (t_k - t)^2 (M_4 + d^2 + d) + d^2(t - t_k)^2 \right) \tag{32}$$

$$\lesssim C_2 d^3 h_k^3 \tag{33}$$

- (30) is derived from (24),

- (31) is derived from Lemma E.2,

- (32) is based on Lemma B.4.

Putting (29) and (33) together, we complete the proof. □

PROOF OF THEOREM 3.1

*Proof.* with Proposition 4.2, Lemma B.2 and Lemma 4.3, we have

$$\mathrm{KL}(p_0 || \hat{q}_T) \leq \mathrm{KL}(p_T || \gamma_d) + T\epsilon_0^2$$

$$+ \sum_{k=1}^{N} \int_{t_{k-1}}^{t_k} \mathbb{E} \| \nabla \log p_{t_k}(x_{t_k}) + \frac{\partial \nabla \log p_{t_k}(x_{t_k})}{\partial t_k} \cdot (t - t_k)$$

$$+ \nabla^2_{x_{t_k}} \log p_{t_k}(x_{t_k}) \cdot (x_t - x_k) - \nabla \log p_t(x_t) \|^2 \mathrm{d}t \tag{34}$$

$$\lesssim (M_2 + d)e^{-T} + T\epsilon_0^2 + \sum_{k=1}^{N} C_2 d^3 h_k^3$$

$$\lesssim (M_2 + d)e^{-T} + T\epsilon_0^2 + \frac{C_2 d^3 T^3}{N^2}$$

□

## C.2 PROOF OF COROLLARY 3.3

The proof of Corollary 3.3 is similar to that of Theorem 3.1. The primary adjustment involves replacing the SDE-DPM-2 scheme, as described by equation (13), with the Runge-Kutta-2 scheme,

detailed in equation (11). This substitution introduces an additional term, $\mathbb{E}\|x_t - x_{t_k}\|^2$, into the calculation of the discretization error. Consequently, the argument supporting the proof of Corollary 3.3 is straightforward and follows logically from this modification.

*Proof.* First, we have the decomposition of the $\mathrm{KL}(p_0||\hat{q}_T)$ as follows:

$$
\mathrm{KL}(p_0||\hat{q}_T) \le \mathrm{KL}\left(p_T||\gamma_d\right) + T\epsilon_0^2
$$

$$
+ \sum_{k=1}^{N} \int_{t_{k-1}}^{t_k} \mathbb{E}\|x_{t_k} - x_t + \nabla \log p_{t_k}(x_{t_k}) + \frac{\partial \nabla \log p_{t_k}(x_{t_k})}{\partial t_k} \cdot (t - t_k) \tag{35}
$$

$$
+ \nabla_{x_{t_k}}^2 \log p_{t_k}(x_{t_k}) \cdot (x_t - x_k) - \nabla \log p_t(x_t)\|^2 \mathrm{d}t
$$

The initial and score estimation errors are identical to those found in SDE-DPM-2, as described by equation 34. Additionally, the term $x_t - x_{t_k}$ in the discretization error originates from the update scheme outlined in equation (11) using the Runge-Kutta-2 method. We then proceed to establish bounds for the discretization error inherent in the Runge-Kutta-2 approach as follows:

$$
\int_{t_{k-1}}^{t_k} \mathbb{E}\|x_{t_k} - x_t + \nabla \log p_{t_k}(x_{t_k}) + \frac{\partial \nabla \log p_{t_k}(x_{t_k})}{\partial t_k} \cdot (t - t_k)
$$

$$
+ \nabla_{x_{t_k}}^2 \log p_{t_k}(x_{t_k}) \cdot (x_t - x_k) - \nabla \log p_t(x_t)\|^2 \mathrm{d}t
$$

$$
\le \int_{t_{k-1}}^{t_k} 2\mathbb{E}\|x_{t_k} - x_t\|^2 \mathrm{d}t + \int_{t_{k-1}}^{t_k} 2\mathbb{E}\|\nabla \log p_{t_k}(x_{t_k}) + \frac{\partial \nabla \log p_{t_k}(x_{t_k})}{\partial t_k} \cdot (t - t_k)
$$

$$
+ \nabla_{x_{t_k}}^2 \log p_{t_k}(x_{t_k}) \cdot (x_t - x_k) - \nabla \log p_t(x_t)\|^2 \mathrm{d}t
$$

$$
\lesssim 2(dh_k^2 + (M_2 + d)h_k^3) + 2C_2 d^3 h_k^3 \tag{36}
$$

The last inequality is derived from Lemma B.1 and Lemma 4.3. Then putting (36) into (35), we complete the proof. □

# D DISCUSSION OF VE-SDE

We will first give Lemma D.1 to bound the expected value of $\|x_t - x_{t_k}\|^2$ and $\|x_t\|^2$ under the VE-SDE. Then we will provide the proof of Corollary 5.1.

**Lemma D.1.** *given the forward process of VE-SDE:*

$$
\mathrm{d}x_t = \sqrt{2}\mathrm{d}W_t \tag{37}
$$

*Under Assumption 1, suppose that $h_k \le 1$ for $1 \le k \le N$, for $t_{k-1} \le t \le t_k$, we have*

$$
\mathbb{E}\|x_t - x_{t_k}\|^2 \lesssim d\left(t_k - t\right)
$$

*and*

$$
\mathbb{E}\|x_t\|^2 \le M_2 + 2d \cdot t
$$

*proof of Lemma D.1.* With (37), we have

$$
\mathbb{E}\|x_t - x_{t_k}\|^2 = \mathbb{E}\|\int_{t_k}^{t} \sqrt{2}\mathrm{d}w_u\|^2
$$

$$
= 2\mathbb{E}\|\int_{t_k}^{t} \mathrm{d}w_u\|^2
$$

$$
\le 2d(t_k - t)
$$

$$
\lesssim d(t_k - t)
$$

we have $x_t|x_0 \sim \mathcal{N}(x_0, 2tI_d)$, then the second moment of $x_t$, $\mathbb{E}\|x_t\|^2$, is bounded by as:

$$
\mathbb{E}\|x_t\|^2 = \mathbb{E}\|x_0 + \sqrt{2}\mathrm{d}w_t\|^2
$$

$$
\le \mathbb{E}\|x_0\|^2 + 2t\mathbb{E}\|w_t\|^2
$$

$$
\le M_2 + 2d \cdot t
$$

The last inequality is derived from Assumption 1. We complete the proof. □

PROOF OF COROLLARY 5.1

With Lemma D.1, we now proceed to prove Corollary 5.1.

When comparing the backward process of VE-SDE, as described in the equation (18) with the backward process of VP-SDE, outlined in the equation referred to as (5), it is notable that the backward process of VE-SDE lacks the $x_t^{\leftarrow}$ term in the drift component. By implementing the same proof strategy as that used in the proof of Corollary 3.3, the discretization errors for both the RK-2 and the SDE-DPM-2 under the VE-SDE framework are found to be identical. Therefore, for the sake of clarity and illustration, we choose to use SDE-DPM-2 as a representative example to explicate the proof of the Corollary 5.1.

*Proof.* similar to Proposition 4.2, we have the decomposition of the $\text{KL}(p_0||\hat{q}_T)$ as follows:

$$
\begin{aligned}
\text{KL}(p_0||\hat{q}_T) \leq \ &\text{KL}\left(p_T||\gamma_d\right) + T\epsilon_0^2 \\
&+ \sum_{k=1}^{N} \int_{t_{k-1}}^{t_k} \mathbb{E}\|\nabla \log p_{t_k}(x_{t_k}) + \frac{\partial \nabla \log p_{t_k}(x_{t_k})}{\partial t_k} \cdot (t - t_k) \\
&\quad + \nabla_{x_{t_k}}^2 \log p_{t_k}(x_{t_k}) \cdot (x_t - x_k) - \nabla \log p_t(x_t)\|^2 \mathrm{d}t
\end{aligned}
\tag{38}
$$

the only difference between the discretization error of VE-SDE and that of VP-SDE is the expected value of $\|x_t - x_{t_k}\|^2$ and $\|x_t\|^2$. Similar to the proof of Lemma 4.3, we have:

$$
\begin{aligned}
\int_{t_{k-1}}^{t_k} &\mathbb{E}\|\nabla \log p_{t_k}(x_{t_k}) + \frac{\partial \nabla \log p_{t_k}(x_{t_k})}{\partial t_k} \cdot (t - t_k) \\
&\quad + \nabla_{x_{t_k}}^2 \log p_{t_k}(x_{t_k}) \cdot (x_t - x_k) - \nabla \log p_t(x_t)\|^2 \mathrm{d}t \\
\leq \int_{t_{k-1}}^{t_k} &(2C_1^2(2\mathbb{E}\|x_{t_k} - x_t\|^2 + 2\mathbb{E}\|x_t\|^2) + 2C_2^2) \cdot h_k^4 \\
&+ \int_{t_{k-1}}^{t_k} C_3^2 \mathbb{E}\|x_t - x_{t_k}\|^4 \mathrm{d}t
\end{aligned}
\tag{39}
$$

Then applying Lemma D.1 to (39), we get

$$
\begin{aligned}
\int_{t_{k-1}}^{t_k} &\mathbb{E}\|\nabla \log p_{t_k}(x_{t_k}) + \frac{\partial \nabla \log p_{t_k}(x_{t_k})}{\partial t_k} \cdot (t - t_k) \\
&\quad + \nabla_{x_{t_k}}^2 \log p_{t_k}(x_{t_k}) \cdot (x_t - x_k) - \nabla \log p_t(x_t)\|^2 \mathrm{d}t \\
&\leq 4C_1^2(dh_k + M_2 + 2d \cdot t)h_k^5 + 2C_2^2 h_k^5 + C_3^2 d^2 h_k^3 \\
&\lesssim C_3^2 d^2 h_k^3
\end{aligned}
\tag{40}
$$

Taking the result of (40) into (38), we finish the proof.

$\square$

# E DISCUSSION OF ASSUMPTIONS UNDER GAUSSIAN MIXTURES

In this section, we will provide that Assumptions 3 and 4 hold for general Gaussian Mixture Model (GMM). Let us consider the general GMM with $K$ components, where the mean and covariance matrix of the $k$-th component are denoted by $\boldsymbol{\mu}_{k,t}$ and $\boldsymbol{\Sigma}_{k,t}$ respectively. We reformulate the assumptions as follows:

**Lemma E.1.** *The second-order derivative of the score function concerning $t$ are bounded, i.e., for all $k = 1, 2, \cdots, N$ and $t \in [t_{k-1}, t_k]$, there exist constants $C_1$ such that:*

$$
\mathbb{E}_{p_t} \left\| \frac{\partial^2 \nabla \log p_t(x)}{\partial t^2} \right\|^2 \leq C_1
$$

*where $C$ is a constant independent of $t$ and only depends on the moments of the initial distribution $p_0$.*

**Lemma E.2.** *The second-order derivative of the score function concerning $x$ are bounded, i.e., for all $k = 1, 2, \cdots, N$ and $t \in [t_{k-1}, t_k]$, there exists a constant $C_2$ such that:*

$$\mathbb{E}_{P_t} \left\| \nabla^3 \log p_t(x) \right\|^2 \leq C_2$$

We first evaluate that Assumptions E.1 and E.2 hold for Gaussian Distribution.

### E.1 GAUSSIAN DISTRIBUTION

Assume the target distribution is Gaussian, i.e., $p_0(x) = \mathcal{N}(x; \mu_0, \Sigma_0)$, where $\Sigma_0$ is a positive definite matrix. Following the forward process,

$$\mathrm{d}x = -x\mathrm{d}t + \sqrt{2}\mathrm{d}w \tag{41}$$

the distribution of $x_t$ at time $t$ is given by

$$x_t \sim \mathcal{N}(x; \mu_t, \Sigma_t) = \mathcal{N}(x; e^{-t}\mu_0, e^{-2t}(\Sigma_0 - I_d) + I_d).$$

Then the score function of $p_t(x)$ is

$$\begin{aligned}
\nabla \log p_t(x) &= \frac{\nabla p_t(x)}{p_t(x)} \\
&= \frac{\mathcal{N}(x|\mu_t, \Sigma_t)\Sigma_t^{-1}(\mu_t - x)}{\mathcal{N}(x|\mu_t, \Sigma_t)} \\
&= \Sigma_t^{-1}(\mu_t - x)
\end{aligned} \tag{42}$$

the second derivative of the score function w.r.t. $t$ is:

$$\begin{aligned}
&\frac{\partial^2 \nabla \log p_t(x)}{\partial^2 t} \\
=&\frac{\partial}{\partial t}\left( \frac{\partial \Sigma_t^{-1}}{\partial t}(\mu_t - x) - \Sigma_t^{-1}\mu_t \right) \\
=&\frac{\partial^2 \Sigma_t^{-1}}{\partial^2 t}(\mu_t - x) - 2\frac{\partial \Sigma_t^{-1}}{\partial t}\mu_t + \Sigma_t^{-1}\mu_t.
\end{aligned} \tag{43}$$

**Lemma E.3.** *the first and second derivative of the inverse covariance matrix of Gaussian distribution w.r.t. $t$ are:*

$$\frac{\partial \Sigma_t^{-1}}{\partial t} = -\Sigma_t^{-1}\left( \frac{\partial \Sigma_t}{\partial t} \right)\Sigma_t^{-1}. \tag{44}$$

$$\frac{\partial^2 \Sigma_t^{-1}}{\partial^2 t} = 2\Sigma_t^{-1}\left( \frac{\partial \Sigma_t}{\partial t} \right)\Sigma_t^{-1}\left( \frac{\partial \Sigma_t}{\partial t} \right)\Sigma_t^{-1} - \Sigma_t^{-1}\left( \frac{\partial^2 \Sigma_t}{\partial^2 t} \right)\Sigma_t^{-1} \tag{45}$$

*where*

$$\frac{\partial \Sigma_t}{\partial t} = -2e^{-2t}\Sigma_0 + 2e^{-2t}I_d,$$

*and*

$$\frac{\partial^2 \Sigma_t}{\partial^2 t} = 4e^{-2t}\Sigma_0 - 4e^{-2t}I_d.$$

*Proof.*

$$\frac{\partial I_d}{\partial t} = \frac{\partial \Sigma_t \Sigma_t^{-1}}{\partial t} = \frac{\partial \Sigma_t}{\partial t}\Sigma_t^{-1} + \Sigma_t \frac{\partial \Sigma_t^{-1}}{\partial t} = 0$$

then we have:

$$\Sigma_t \frac{\partial \Sigma_t^{-1}}{\partial t} = -\frac{\partial \Sigma_t}{\partial t}\Sigma_t^{-1}$$

left multiply $\Sigma_t^{-1}$ on both sides, we get:

$$\frac{\partial \Sigma_t^{-1}}{\partial t} = -\Sigma_t^{-1} \frac{\partial \Sigma_t}{\partial t} \Sigma_t^{-1}$$

with the same method, we can also proof Eq.(45).

Since we have $\Sigma_t = e^{-2t}(\Sigma_0 - I_d) + I_d$, we can directly get

$$\frac{\partial \Sigma_t}{\partial t} = -2e^{-2t}\Sigma_0 + 2e^{-2t}I_d,$$

and

$$\frac{\partial^2 \Sigma_t}{\partial^2 t} = 4e^{-2t}\Sigma_0 - 4e^{-2t}I_d.$$

Thus we complete the proof. $\square$

Denote the eigenvalues of $\Sigma_0$ as $\lambda_1 \leq \lambda_2 \leq \ldots \leq \lambda_d$.

Then we have the largest and smallest eigenvalues of $\Sigma_t$ as:

$$\lambda_{\min}(\Sigma_t) = e^{-2t}(\lambda_{\min} - 1) + 1 = \min(1, \lambda_1)$$

$$\lambda_{\max}(\Sigma_t) = e^{-2t}(\lambda_{\max} - 1) + 1 = \max(1, \lambda_d).$$

**Lemma E.4.** *Assume $p_0(x) = \mathcal{N}(x; \mu_0, \Sigma_0)$ is a Gaussian distribution with $\Sigma_0$ being a general positive definite matrix, with eigenvalues $\lambda_1 \leq \lambda_2 \leq \ldots \leq \lambda_d$. We have the following bounds for the second derivative of the score function w.r.t. $t$:*

$$\left\| \frac{\partial^2 \nabla \log p_t(x)}{\partial^2 t} \right\| \leq C_1 \|x\| + C_2 \tag{46}$$

*with*

$$C_1 = \left( \frac{8(\lambda_d + 1)^2}{\min(1, \lambda_{\min}^3(\Sigma_0))} + \frac{4(\lambda_d + 1)}{\min(1, \lambda_{\min}^2(\Sigma_0))} \right)$$

*and*

$$C_2 = \left( \frac{8(\lambda_d + 1)^2}{\min(1, \lambda_{\min}^3(\Sigma_0))} + \frac{4(\lambda_d + 1)}{\min(1, \lambda_{\min}^2(\Sigma_0))} \right.$$
$$\left. + \frac{4(\lambda_d + 1)}{\min(1, \lambda_{\min}^2(\Sigma_0))} + \frac{1}{\min(1, \lambda_1)} \right) \|\mu_0\|$$

*and the bound for the second derivative of the score function w.r.t. $x$:*

$$\mathbb{E} \left\| \frac{\partial^2 \nabla \log p_t(x)}{\partial x^2} \right\| = 0$$

*Proof.* First, we need to bound the largest eigenvalues of $\Sigma_t^{-1}$. The largest eigenvalue of $\Sigma_t^{-1}$ is directly bounded by

$$\lambda_{\max}(\Sigma_t^{-1}) = \frac{1}{\lambda_{\min}(\Sigma_t)} = \frac{1}{\min(\lambda_1, 1)}$$

from Eq.(43), we get

$$\frac{\partial^2 \nabla \log p_t(x)}{\partial^2 t} = \left( 2\Sigma_t^{-1} \left( \frac{\partial \Sigma_t}{\partial t} \right) \Sigma_t^{-1} \left( \frac{\partial \Sigma_t}{\partial t} \right) \Sigma_t^{-1} \right.$$
$$\left. - \Sigma_t^{-1} \left( \frac{\partial^2 \Sigma_t}{\partial^2 t} \right) \Sigma_t^{-1} \right) (\mu_t - x)$$
$$+ 2\Sigma_t^{-1} \left( \frac{\partial \Sigma_t}{\partial t} \right) \Sigma_t^{-1} \mu_t + \Sigma_t^{-1} \mu_t$$

We have the following bound for the second derivative of the score function w.r.t. $t$:

$$\|\frac{\partial^2 \nabla \log p_t(x)}{\partial^2 t}\|$$

$$= \| \left( 2\Sigma_t^{-1} \left(\frac{\partial \Sigma_t}{\partial t}\right) \Sigma_t^{-1} \left(\frac{\partial \Sigma_t}{\partial t}\right) \Sigma_t^{-1} \right.$$

$$- \Sigma_t^{-1} \left(\frac{\partial^2 \Sigma_t}{\partial^2 t}\right) \Sigma_t^{-1} \right) (\mu_t - x)$$

$$+ 2\Sigma_t^{-1} \left(\frac{\partial \Sigma_t}{\partial t}\right) \Sigma_t^{-1} \mu_t + \Sigma_t^{-1} \mu_t \|$$

$$\underbrace{\leq}_{(i)} \left(\frac{8(\lambda_d + 1)^2}{\lambda_{\min}^3(\Sigma_t)} + \frac{4(\lambda_d + 1)}{\lambda_{\min}^2(\Sigma_t)}\right) \|\mu_t - x\|$$

$$+ \left(\frac{4(\lambda_d + 1)}{\lambda_{\min}^2(\Sigma_t)} + \frac{1}{\lambda_{\min}(\Sigma_t)}\right) \|\mu_t\|$$

$$\underbrace{\leq}_{(ii)} \left(\frac{8(\lambda_d + 1)^2}{\lambda_{\min}^3(\Sigma_t)} + \frac{4(\lambda_d + 1)}{\lambda_{\min}^2(\Sigma_t)}\right) \|x\|$$

$$+ \left(\frac{8(\lambda_d + 1)^2}{\lambda_{\min}^3(\Sigma_t)} + \frac{4(\lambda_d + 1)}{\lambda_{\min}^2(\Sigma_t)}\right.$$

$$\left. + \frac{4(\lambda_d + 1)}{\lambda_{\min}^2(\Sigma_t)} + \frac{1}{\lambda_{\min}(\Sigma_t)}\right) \|\mu_0\| \qquad (47)$$

where $(i)$ is from the triangle inequality, and the scaling of Matrix Spectral Norm; and $(ii)$ is due to $\|\mu_t\| = e^{-t}\|\mu_0\| \leq \|\mu_0\|$. With $\lambda_{\min}(\Sigma_t) = \min(1, \lambda_1)$.

As for the derivative of the score function w.r.t. $x$, according to Eq.(42), we directly get

$$\frac{\partial \nabla \log p_x(x)}{\partial x} = \Sigma_t^{-1}$$

The second derivative of the score function w.r.t. $x$ is $\mathbf{0}$. We complete the proof. $\qquad \square$

### E.2 GAUSSIAN MIXTURE DISTRIBUTION

#### E.2.1 DERIVATIVES OF THE SCORE FUNCTION W.R.T. $t$

Let us first consider the most simple case:

$$p_0(x) = \pi_1 \mathcal{N}(x; \mu_1, I_d) + \pi_2 \mathcal{N}(x :, \mu_2, I_d)$$

, where $\pi_1 + \pi_2 = 1$, and $\pi_1, \pi_2 > 0$.

Let $p_1 = \mathcal{N}(x; \mu, I_d)$, and $p_2 = \mathcal{N}(x; \mu_2, I_d)$ and $\pi_1 = \pi_2 = \frac{1}{2}$, we have

$$\nabla \log p_t(x) = \frac{p_1(\mu_1 - x) + p_2(\mu_2 - x)}{p_1 + p_2}$$

the first derivative w.r.t. $t$ is

$$\frac{\partial \nabla \log p_t(x)}{\partial t}$$

$$= \frac{p_1^2 \mu_1' + p_2^2 \mu_2' + p_1 p_2 (\mu_1' + \mu_2') \left(1 + (\mu_1 - x)(\mu_2 - x)\right)}{(p_1 + p_2)^2}$$

$$- \frac{p_1 p_2 \left((\mu_1 - x)^2 \mu_1' + (\mu_2 - x)^2 \mu_2'\right)}{(p_1 + p_2)^2}$$

with the fact that $\mu_1' = -\mu_1$, and $\mu_2' = -\mu_2$, we rearrange the terms and get

$$
\frac{\partial \nabla \log p_t(x)}{\partial t}
$$

$$
= \frac{p_1 p_2 \left( (\mu_1 + \mu_2)((\mu_1 - \mu_2)^2 - 1) - (\mu_1 - \mu_2)^2 x \right)}{(p_1 + p_2)^2}
$$

$$
- \frac{p_1^2 \mu_1 + p_2^2 \mu_2}{(p_1 + p_2)^2}
$$

$$
= \frac{p_1 p_2 \left( (\mu_1 - \mu_2)^2 (\mu_1 + \mu_2 - x) \right)}{(p_1 + p_2)^2} - \frac{p_1 \mu_1 + p_2 \mu_2}{p_1 + p_2}
$$

with the fact that

$$
\frac{\partial p_i}{\partial t} = p_i \mu_i (\mu_i - x)
$$

*it seems the second derivative $\frac{\partial^2 \nabla \log p_t(x)}{\partial^2 t}$ depend on $x$ and $x^2$.*

Let us first evaluate the case for two components Gaussian Mixture Distribution:

assume $p_1(x) = \mathcal{N}(x; \mu_i, \Sigma_i) = \frac{1}{(2\pi)^{n/2} \det \Sigma_i} e^{-\frac{1}{2}(x - \mu_1)^\top \Sigma_1^{-1}(x - \mu_1)}$, $i = 1, \ldots, K$ we have the following facts:

$$
\nabla \log p(x) = \frac{\pi_1 p_1 \Sigma^{-1}(\mu_1 - x) + \pi_2 p_2 \Sigma^{-1}(\mu_2 - x)}{\pi_1 p_1 + \pi_2 p_2}
$$

$$
p_i' = \frac{\partial p_i}{\partial t} = p_i \left( \mu_i^\top \Sigma^{-1}(\mu_i - x) - \frac{1}{2} \mathrm{tr}\left( (\mu_i - x)(\mu_i - x)^\top \frac{\partial \Sigma^{-1}}{\partial t} \right) \right.
$$

$$
\left. - \frac{1}{2} p_i \frac{\mathrm{tr}\left( \Sigma^{-1} \frac{\partial \Sigma}{\partial t} \right)}{\sqrt{\det(\Sigma)}} \right)
$$

$$
p_i' = p_i \left( \mu_i^\top \Sigma^{-1}(\mu_i - x) + (\mu_i - x)^\top \left( \Sigma^{-1}(\Sigma^{-1} - I_d) \right)(\mu_i - x) \right. \tag{48}
$$

$$
\left. - \frac{1}{2} p_i \frac{\mathrm{tr}\left( \Sigma^{-1} \frac{\partial \Sigma}{\partial t} \right)}{\sqrt{\det(\Sigma)}} \right) = p_i(u_{i,1} + u_{i,2} + u_{i,3})
$$

$$
p_i'' = \frac{\partial^2 p_i}{\partial^2 t} = \frac{\partial p_i}{\partial t}(u_{i,1} + u_{i,2} + u_{i,3}) + p_i \left( \frac{\partial u_{i,1}}{\partial t} + \frac{\partial u_{i,2}}{\partial t} + \frac{\partial u_{i,3}}{\partial t} \right)
$$

$$
= \frac{\partial p_i}{\partial t}(u_{i,1} + u_{i,2} + + u_{i,3})
$$

$$
+ p_i \left( \mu_i^\top \Sigma^{-1}(x - 2\mu_i) + \mathrm{tr}\left( (\mu_i - x)\mu_i^\top \frac{\partial \Sigma^{-1}}{\partial t} \right) \right)
$$

$$
+ p_i \, \mathrm{tr}\left( -\left( \mu_i(\mu_i - x)^\top + (\mu_i - x)\mu_i^\top \right) \frac{\partial \Sigma^{-1}}{\partial t} \right)
$$

$$
+ p_i \, \mathrm{tr}\left( (\mu_i - x)(\mu_i - x)^\top \frac{\partial^2 \Sigma^{-1}}{\partial^2 t} \right)
$$

$$
+ p_i \left( -\frac{1}{2} \frac{\mathrm{tr}\left( \frac{\partial \Sigma_i^{-1}}{\partial t} \frac{\partial \Sigma_i}{\partial t} + \Sigma^{-1} \frac{\partial^2 \Sigma_i}{\partial^2 t} \right)}{\sqrt{\det(\Sigma_i)}} + \frac{1}{4} \frac{\left( \mathrm{tr}\left( \Sigma^{-1} \frac{\partial \Sigma}{\partial t} \right) \right)^2}{\sqrt{\det(\Sigma)}} \right)
$$

Thus, for $\Sigma_i^{-1} \neq I_d$, we have

$$
\frac{\partial \nabla \log p_t(x)}{\partial t} = \frac{(p_2 p_1' - p_1 p_2')(S_1 - S_2)}{(p_1 + p_2)^2}
$$

$$
+ \frac{p_1 S_1' + p_2 S_2'}{p_1 + p_2}
$$

$$\frac{\partial^2 \nabla \log p_t(x)}{\partial^2 t} = \frac{\partial}{\partial t}\left(\frac{(p_2 p_1' - p_1 p_2')(S_1 - S_2)}{(p_1 + p_2)^2}\right)$$

$$+ \frac{\partial}{\partial t}\left(\frac{p_1 S_1' + p_2 S_2'}{p_1 + p_2}\right)$$

$$= part1 + part2$$

in the same method, we get

$$part1 = \frac{(p_2 p_1' - p_1 p_2')(S_1' - S_2')}{(p_1 + p_2)^2}$$

$$+ \frac{(p_2 p_1'' - p_1 p_2'')(S_1 - S_2)}{(p_1 + p_2)^2}$$

$$- 2\frac{(p_2 p_1' - p_1 p_2')(S_1 - S_2)(p_1' + p_2')}{(p_1 + p_2)^3}$$

$$part2 = \frac{p_1 S_1'' + p_2 S_2''}{p_1 + p_2} + \frac{(p_2 p_1' - p_1 p_2')(S_1' - S_2')}{(p_1 + p_2)^2}$$

- $\Sigma \neq I_d$, $p''$ will involve $\|x\|^3$.
- $\Sigma = I_d$, we have the following bound:

  the first derivative of the score function w.r.t. $t$ is

  $$\mathbb{E}\left\|\frac{\partial \log p_t(x)}{\partial t}\right\|$$

  $$\leq \mathbb{E}\left\|\left(\mu_1^\top \Sigma^{-1}(\mu_1 - x) - \mu_2^\top \Sigma^{-1}(\mu_2 - x)\right)\left(\Sigma^{-1}(\mu_1 - \mu_2)\right)\right\|$$

  $$+ \mathbb{E}\|\mu_1 - \mu_2\|$$

  $$= \mathbb{E}\left\|\left(\mu_1^\top(\mu_1 - x) - \mu_2^\top(\mu_2 - x)\right)(\mu_1 - \mu_2)\right\| + \|\mu_1 - \mu_2\|$$

  $$\leq \left(\|\mu_1\|^2 + \|\mu_2\|^2 + \|\mu_1 - \mu_2\|\mathbb{E}\|x\|\right)\|\mu_1 - \mu_2\| + \|\mu_1 - \mu_2\|$$

  which is the form of $C_1\|x\| + C_2$.

  the second derivative of the score function w.r.t. $t$ is

  $$\mathbb{E}\left\|\frac{\partial^2 \log p_t(x)}{\partial^2 t}\right\|$$

  $$\leq \mathbb{E}\left\|\frac{p_1'}{p_1} - \frac{p_2'}{p_2}\right\|\mathbb{E}\|S_1' - S_2'\| + \mathbb{E}\left\|\frac{p_1''}{p_1} - \frac{p_2''}{p_2}\right\|\|S_1 - S_2\|$$

  $$+ \mathbb{E}\left\|\frac{p_1'}{p_1} - \frac{p_2'}{p_2}\right\|\|S_1 - S_2\|\left\|\frac{p_1'}{p_1} + \frac{p_2'}{p_2}\right\|$$

  $$+ \mathbb{E}\|S_1''\| + \mathbb{E}\|S_2''\|$$

  which is the form of $C_1\|x\|^2 + C_2\|x\| + C_3$.

### E.2.2 DERIVATIVES OF THE SCORE FUNCTION W.R.T. $x$

Now we come to the bound of the derivative of the score function w.r.t. $x$.

The first derivative of the score function w.r.t. $x$ for $p(x) = \pi_1 \mathcal{N}(x; \mu_1, I_d) + \pi_2 \mathcal{N}(x; \mu_2, I_d)$, is

$$\frac{\partial \nabla \log p_t(x)}{\partial x}$$

$$= -1 + \frac{\pi_1 \pi_2 p_1 p_2 \left((\mu_1 - x)^2 + (\mu_2 - x)^2 - 2(\mu_1 - x)(\mu_2 - x)\right)}{(\pi_1 p_1 + \pi_2 p_2)^2}$$

$$= -1 + \frac{\pi_1 \pi_2 p_1 p_2 (\mu_1 - \mu_2)^2}{(\pi_1 p_1 + \pi_2 p_2)^2}$$

The second derivative of the score function w.r.t. $x$ is

$$\frac{\partial^2 \nabla \log p_t(x)}{\partial x^2} = \frac{\partial \frac{\partial \nabla \log p_t(x)}{\partial x}}{\partial x}$$

$$= -\frac{(\mu_1 - \mu_2)^3 p_1 p_2 (p_1 - p_2)}{(p_1 + p_2)^3}$$

if $\Sigma_1, \Sigma_2$ are not identity matrices, the first derivative of the score function w.r.t. $x$ is as follows:

$$\frac{\partial \nabla \log p_t(x)}{\partial x}$$

$$= -\frac{p_1 \Sigma_1^{-1} + p_2 \Sigma_2^{-1}}{(p_1 + p_2)} + \frac{p_1 p_2 (S_1 - S_2)(S_1 - S_2)^\top}{(p_1 + p_2)^2}$$

the second derivative of the score function w.r.t. $x$ is (verified on 1-d case):

$$
\begin{aligned}
&\frac{\partial \frac{\partial \nabla \log p_t(x)}{\partial x}}{\partial x} \\
&= -\frac{p_1 p_2}{(p_1 + p_2)^2} \left( \text{Vec} \left( \Sigma_1^{-1} - \Sigma_2^{-1} \right) \otimes (S_1 - S_2)^\top \right. \\
&\quad + (\Sigma_1^{-1} - \Sigma_2^{-1}) \otimes (S_1 - S_2) + (S_1 - S_2) \otimes (\Sigma_1^{-1} - \Sigma_2^{-1})) \\
&\quad - \frac{p_1 p_2 (p_1 - p_2)}{(p_1 + p_2)^3} \text{Vec} \left( (S_1 - S_2)(S_1 - S_2)^\top \right) \otimes (S_1 - S_2)^\top
\end{aligned}
\tag{49}
$$

where $S_k = \Sigma_k^{-1}(\mu_k - x)$, $k = 1, 2$ and $\text{Vec}(A)$ involves concatenating the columns of the matrix $A$ sequentially to form a single column vector.

**Lemma E.5.** *Assume the target distribution is a 2-component Gaussian Mixture Distribution on $\mathbb{R}^1$, and denote $p_0(x) = \frac{1}{2}\mathcal{N}(x; \mu_1, \sigma_1) + \frac{1}{2}\mathcal{N}(x; \mu_2, \sigma_2)$, let $\delta_\sigma$ be absolute difference of $\sigma_1$ and $\sigma_2$, i.e., $\delta_\sigma = |\sigma_1 - \sigma_2|$, we have the following bounds for the second derivative of the score function w.r.t. $x$:*

$$\mathbb{E} \left\| \frac{\partial^2 \nabla \log p_t(x)}{\partial^2 x} \right\|^2 \leq C_3 \tag{50}$$

*where*

$$
\begin{aligned}
C_3 =& 2 \frac{e^{-8t \delta_\sigma}}{\sigma_{1,t}^8 \sigma_{2,t}^8} M_{2,t} + \frac{e^{-16t \delta_\sigma}}{\sigma_{1,t}^{12} \sigma_{2,t}^{12}} M_{6,t} + \left( \frac{\sigma_{1,t}^2 \mu_{2,t} + \sigma_{2,t}^2 \mu_{1,t}^3}{\sigma_{1,t}^6 \sigma_{2,t}^6} \right)^6 \\
&+ \left( \frac{p_1 p_2 (\frac{1}{\sigma_{1,t}^2} - \frac{1}{\sigma_{2,t}^2})(\frac{\mu_{1,t}}{\sigma_{1,t}^2} - \frac{\mu_{2,t}}{\sigma_{2,t}^2})}{(p_1 + p_2)^2} \right)^2
\end{aligned}
$$

*$M_{2,t}$ and $M_{6,t}$ are the second and sixth moments of the target distribution at time $t$, and $\mu_{k,t} = e^{-t}\mu_k$, and $\sigma_{k,t} = \sqrt{e^{-2t}\sigma_k^2 + 1 - e^{-2t}}$, $k = 1, 2$.*

*Proof.* From Eq.(49), we have

$$
\begin{aligned}
&\frac{\partial^2 \nabla \log p_t(x)}{\partial^2 x} \\
&= \frac{p_1 p_2 (\frac{1}{\sigma_{1,t}^2} - \frac{1}{\sigma_{2,t}^2})^2 x}{(p_1 + p_2)^2} \\
&\quad - \frac{p_1 p_2 (\frac{1}{\sigma_{1,t}^2} - \frac{1}{\sigma_{2,t}^2})(\frac{\mu_{1,t}}{\sigma_{1,t}^2} - \frac{\mu_{2,t}}{\sigma_{2,t}^2})}{(p_1 + p_2)^2} \\
&\quad + \frac{\left( (\sigma_{2,t}^2 - \sigma_{1,t}^2)x + \sigma_{1,t}^2 \mu_{2,t} - \sigma_{2,t}^2 \mu_{1,t} \right)^3 p_1 p_2 (p_1 - p_2)}{\sigma_{1,t}^6 \sigma_{2,t}^6 (p_1 + p_2)^3}
\end{aligned}
$$

Thus we get

$$\mathbb{E}_{p_t} \left\| \frac{\partial^2 \nabla \log p_t(x)}{\partial^2 x} \right\|^2$$

$$\leq \underbrace{\mathbb{E} \left\| \frac{p_1 p_2 (\frac{1}{\sigma_{1,t}^2} - \frac{1}{\sigma_{2,t}^2})^2 x}{(p_1 + p_2)^2} \right\|^2}_{term \; I}$$

$$+ \underbrace{\mathbb{E} \left( \frac{p_1 p_2 (\frac{1}{\sigma_{1,t}^2} - \frac{1}{\sigma_{2,t}^2})(\frac{\mu_{1,t}}{\sigma_{1,t}^2} - \frac{\mu_{2,t}}{\sigma_{2,t}^2})}{(p_1 + p_2)^2} \right)^2}_{term \; II}$$

$$+ \underbrace{\mathbb{E} \left\| \frac{\left( (\sigma_{2,t}^2 - \sigma_{1,t}^2) x + \sigma_{1,t}^2 \mu_{2,t} - \sigma_{2,t}^2 \mu_{1,t} \right)^3 p_1 p_2 (p_1 - p_2)}{\sigma_{1,t}^6 \sigma_{2,t}^6 (p_1 + p_2)^3} \right\|^2}_{term \; III}$$

for *term I*, we have

$$term \; I \leq (|\frac{1}{\sigma_{1,t}^2} - \frac{1}{\sigma_{2,t}^2}|)^4 \mathbb{E} \|x\|^2$$

$$\leq 2 \frac{e^{-8t} \delta_\sigma}{\sigma_{1,t}^8 \sigma_{2,t}^8} \mathbb{E} \|x\|^2$$

$$\underbrace{=}_{(i)} 2 \frac{e^{-8t} \delta_\sigma}{\sigma_{1,t}^8 \sigma_{2,t}^8} M_{2,t}$$

for *term II*, it does not depend on $x$, we directly get

$$term \; II = (\frac{p_1 p_2 (\frac{1}{\sigma_{1,t}^2} - \frac{1}{\sigma_{2,t}^2})(\frac{\mu_{1,t}}{\sigma_{1,t}^2} - \frac{\mu_{2,t}}{\sigma_{2,t}^2})}{(p_1 + p_2)^2})^2$$

for *term III*, we have

$$term \; III \leq \mathbb{E} \left\| \frac{\left( (\sigma_{2,t}^2 - \sigma_{1,t}^2) x + \sigma_{1,t}^2 \mu_{2,t} - \sigma_{2,t}^2 \mu_{1,t} \right)^3}{\sigma_{1,t}^6 \sigma_{2,t}^6} \right\|^2$$

$$\leq \left( \frac{\sigma_{1,t}^2 \mu_{2,t} + \sigma_{2,t}^2 \mu_{1,t}^3}{\sigma_{1,t}^6 \sigma_{2,t}^6} \right)^6$$

$$+ \frac{e^{-16t} \delta_\sigma}{\sigma_{1,t}^{12} \sigma_{2,t}^{12}} M_{6,t}$$

$$\qquad \qquad \qquad \qquad \qquad \qquad \qquad \qquad \qquad \qquad \qquad \qquad \qquad \qquad \square$$

Although we analyze the case where the number of components is 2, the results can be easily extended to the case where the number of components is $K$, $p_0(x) = \sum_{k=1}^{K} \pi_k \mathcal{N}(x; \mu_k, \Sigma_k)$.

**Lemma E.6.** *Assume $p_0 = \sum_{k=1}^{K} \pi_k \mathcal{N}(x; \mu_k, \Sigma_k)$ is a Gaussian Mixture Distribution, and denote $p_t(x) = \sum_{k=1}^{K} \pi_k \mathcal{N}(x; \mu_{k,t}, \Sigma_{k,t}) = \sum_{k=1}^{K} \pi_k p_k$ along the forward process (41), we have the following results for the score of $p_t$ and its derivatives: the score function of $p_t$ is*

$$\nabla \log p_t(x) = \frac{\sum_{k=1}^{K} \pi_k p_k \left( \Sigma_{k,t}^{-1} (\mu_{k,t} - x) \right)}{\sum_{i=j}^{K} \pi_k p_k}$$

$$= \frac{\sum_{k=1}^{K} \pi_k p_k S_{k,t}}{\sum_{k}^{K} \pi_k p_k}$$

*the derivative of the score function w.r.t. $x$ is*

$$\frac{\partial \nabla \log p_t(x)}{\partial x} = -\frac{\sum_{k=1}^{K} \pi_k p_k \Sigma_{k,t}^{-1}}{\sum_{k=1}^{K} \pi_k p_k}$$
$$+ \frac{\sum_{k=1}^{K} \sum_{j=k+1}^{K} \pi_k \pi_j p_k p_j (S_{k,t} - S_{j,t})^{\otimes 2}}{(\sum_{k=1}^{K} \pi_k p_k)^2}$$

*the second derivative of the score function w.r.t. $x$ is*

$$\frac{\partial^2 \nabla \log p_t(x)}{\partial x^2}$$
$$= -\frac{\sum_{k=1}^{K} \sum_{j=k+1}^{K} \pi_k \pi_j p_k p_j \operatorname{Vec}\left(\Sigma_{k,t}^{-1} - \Sigma_{j,t}^{-1}\right) \otimes (S_{k,t} - S_{j,t})^\top}{(\sum_{k=1}^{K} \pi_k p_k)^2}$$
$$- \frac{1}{(\sum_{k=1}^{K} \pi_k p_k)^2} \left( \sum_{k=1}^{K} \sum_{j=k+1}^{K} \pi_k \pi_j p_k p_j \cdot \right.$$
$$\left. \left((\Sigma_{k,t}^{-1} - \Sigma_{j,t}^{-1}) \otimes (S_{k,t} - S_{j,t}) + (S_{k,t} - S_{j,t}) \otimes (\Sigma_{k,t}^{-1} - \Sigma_{j,t}^{-1})\right)\right)$$
$$+ \frac{\sum_{k=1}^{K} \sum_{j=k+1}^{K} \pi_k p_k \pi_j p_j \operatorname{Vec}\left((S_{k,t} - S_{j,t})(S_{k,t} - S_{j,t})^\top\right) \otimes C^\top}{(\sum_{k=1}^{K} \pi_k p_k)^3}$$

*where*

$$C = (\pi_j p_j - \pi_k p_k)(S_{k,t} - S_{j,t})$$
$$+ \sum_{h \neq k,j} \pi_h p_h (S_{k,t} + S_{j,t} - 2S_{h,t})$$

With Lemma E.6, the proof for $K-$component Gaussian Mixture Distribution is similar to the 2-component case.