# OpenReview forum: "The Convergence of Second-Order Sampling Methods for Diffusion Models"
_ICLR.cc/2025/Conference — Submitted to ICLR 2025_

### Official Review · Reviewer_4rK4 · 2024-10-27

**Soundness:** 2
**Presentation:** 2
**Contribution:** 2
**Rating:** 5
**Confidence:** 4

**Summary:**

This paper investigates the convergence properties of a second-order discretization method, SDE-DPM-2, for diffusion models. The main result demonstrates that SDE-DPM-2 achieves an improved $O(1/\epsilon)$ convergence rate to obtain an $O(\epsilon^2)$ error in KL divergence, surpassing the performance of existing EI discretization methods. Additionally, using similar proof techniques, the paper shows that another widely used second-order method, Runge-Kutta, does not attain this level of convergence. Further analysis extends these results to the VE SDE, achieving a comparable convergence rate.

**Strengths:**

1. The writing is very clear. It provides both theoretical and empirical comparisons with the most related papers.

2. It proves a better convergence rate for a second-order sampling method.

3. It also extends the setting to VE SDEs, showing that the analysis framework can be further generalized.

**Weaknesses:**

(1) The biggest weakness of this paper is the stringent assumptions. In Assumption 2, this paper assumes the Taylor expansion is accurate, while in most of the previous works for SDE analysis, only the value accuracy is needed. I have seen similar assumptions in [1], which assume the closeness of the Jacobian matrix with respect to $x$ in dealing with ODE analysis. They also showed that such an assumption is not required for SDE. However, Assumption 2 in this paper, though for SDE analysis, is even stronger than that, because it assumes that the time-derivative is also close.

Moreover, Assumptions 3 and 4 are also very strong. When t is close to 0, the score function will get close to the gradient of the probability density function of the data distribution. Such boundness assumptions thus will require the smoothness of the data distribution. As is shown in Appendix B, it can only hold when the data distribution is a Gaussian mixture. This diverges from many useful data distributions, especially when the data is constrained on a low-dimension manifold. As a result, I think more discussions are required to verify the reasonability of these assumptions.

(2) The writing of the paper is a little inconsistent. For example, in equation (6), the first-order derivative is approximated with the value of the score function, while in equation (13) it becomes the partial derivative concerning t and x. Moreover, the notation used here, defined in Line 204 is not standard and very confusing. In Line 201, it says “The difference between the EI and SDE-DPM-2 schemes lies in the approximation of the score function”, while in Line 401, it says “The key difference between EI and SDE-DPM-2 lies in the update scheme at each time interval” It is unclear whether they have the same meaning or not.

(3) The description of the contribution is a little bit inaccurate. It claims that SDE-DPM-2 is more efficient than Runge Kutta. However, no guarantee has been given (Corollary 3.3 only shows that the method used in this paper cannot provide a better guarantee for Runge Kutta). It is possible that there exists an analysis of Runge Kutta that can achieve better results.  As is shown in the experiment, the performance of Runge Kutta and SDE-DPM-2 is similar, both better than first-order methods. Thus, the claim seems a little strange to me. Moreover, this paper says that for VE SDE, the convergence is aligned with VP SDE. However, the remark under Corollary 5.1 shows that it only works when overlooking the initial error, which is the key difficulty of VE SDE. This point should also be emphasized in the introduction.

(4) The paper is not self-contained. For example, the proof of Proposition 4.2, directly refers to Chen et al 2023a without any explanation. In my opinion, the argument here is far from trivial and should not be omitted.

----
[1] Li et al. 2024 Towards Non-Asymptotic Convergence for Diffusion-based Generative Models ICLR2024

**Questions:**

Is it possible to get a better convergence rate for Runge Kutta methods?

---

> ### Author Response · Authors · 2024-11-21
>
> # Response to Weakness:
> - **[assumption 2]**: We acknowledge that Assumption 2 may appear stringent compared to previous works. The reason we introduce this assumption is tied to the specific algorithm we are analyzing: SDE-DPM-2. At each update step, this algorithm requires the first derivative of the score function, as described above in equation (6). This naturally leads to a requirement for a close estimation of the score’s first derivative. Furthermore, as discussed under Assumption 2, Meng et al. (2021) empirically demonstrated that the first derivative of the score can be learned effectively under Gaussian mixture models. Therefore, despite the stringency of Assumption 2, our approach remains empirically reasonable, supported by experimental results.
> - **[assumptions 3 and 4]**: We would like to clarify that in Appendix B, we specifically verify that Assumptions 3 and 4 hold in the case of a Gaussian mixture distribution. For other types of distributions, such as those supported on a low-dimensional manifold, it is currently unclear whether these assumptions hold as $t$ approaches 0. This is something we hope to explore further in future work.
> - **[inconsistent]**: We would like to clarify that in equation (6), the first-order derivative represents the total derivative of the score function with respect to $t$.  As explained below equation (12), we use the chain rule to express total derivatives in terms of partial derivatives. Regarding the statements in Lines 201 and 401, both refer to the same concept: the key difference between EI and SDE-DPM-2 lies in how the score function is approximated within the update scheme at each time step. We have revised the manuscript to ensure that this distinction is more clearly explained and standardize the notation to avoid any confusion.
> - **[rk-2]**: We apologize for any misunderstanding. Corollary 3.3 indeed only indicates that, under our current analytical method, Runge-Kutta does not demonstrate higher efficiency than SDE-DPM-2.
>
>   Regarding the experimental results: we have redrawn the comparison of KL divergence against the number of discretization steps on a log scale. While RK-2 achieves a slightly lower KL divergence than SDE-DPM, the rate of decrease for both methods is similar and slower than that of SDE-DPM-2. This observation aligns with our theoretical findings.
>   Moreover, we acknowledge that there is a gap between existing theoretical analyses and empirical observations, as experiments show that the KL divergence for each method decreases more rapidly than the theoretical bounds. This suggests potential directions for future research to further improve the convergence rate of these methods.
> - **[VE]**: In VP-type diffusion models, the initial error term decays exponentially fast, thus contributing minimally to overall convergence. On the other hand, as shown in equation (17), the VE-type diffusion model has an initial error term that decays at a polynomial rate, specifically $1/T$, which becomes the dominant factor relative to the discretization error term of $1/T^2$. Therefore, in the case of the VE-type diffusion model, when the initial error is overlooked (i.e., assuming the reverse process distribution $q_0$ at time $0$ matches the forward process distribution $p_T$ at time $T$), we can directly compare the algorithm’s discretization error. This key point has been emphasized in the introduction of the revised manuscript.
> - **[Proposition 4.2]**: We have provided a more detailed explanation of the proof of Proposition 4.2 in  Appendix C.1 of the revised manuscript to ensure that the argument is clear and self-contained.
> #  Response to Question:
> - **[ better convergence rate for rk-2]**: Our analysis focuses on the stochastic RK-2 algorithm described in Equations (9-10). The claim regarding the lower efficiency of RK-2 is primarily based on its direct discretization of the linear component of the drift term in the SDE. If the error bound for this discretization could be further reduced, it might be possible to achieve a better convergence rate for Runge-Kutta methods.

---

> > ### Comment · Reviewer_4rK4 · 2024-11-23
> > **Response to authors**
> >
> > Thanks for your response and clarification. After reading it, I find that the analysis in this paper is meaningful, which is interesting to this research community. However, the weaknesses that I mentioned do exist. It weakens the value of this work. As a result, I think 5 is a reasonable score, and I will keep my score.
> >
> > Besides, as mentioned by other reviewers, such as xHXz and xXao, this paper lacks comparison with many related works. I'm curious about their responses to the authors' rebuttal.

---

### Official Review · Reviewer_xHXz · 2024-10-30

**Soundness:** 3
**Presentation:** 2
**Contribution:** 2
**Rating:** 3
**Confidence:** 4

**Summary:**

The paper investigates the convergence of the second-order discretization method (SDE-DPM-2). Given an $O(\epsilon^2)$ $L^2$-accurate score estimation, the paper demonstrates that the sampling complexity of SDE-DPM-2 is $O(1/\epsilon)$ instead of that of the exponential integrator scheme, which is $O(1/\epsilon^2)$. Furthermore, the paper extends the analysis to the Runge-Kutta-2 (RK-2) method, proving that SDE-DPM-2 exhibits superior efficiency compared to RK-2.

**Strengths:**

- The paper studies the SDE-DPM-2 scheme for the inference of diffusion models and improves the sample complexity from $O(1/\epsilon^2)$ to $O(1/\epsilon)$.
- The mathematical proof looks sound to me.
- Several experiments are conducted to validate the theoretical findings.

**Weaknesses:**

- The assumptions appear overly strong and artificial to me. Unlike the conventional assumption that the neural network score function $s(t, \cdot)$ is approximately $\epsilon^2$ close to the true score function $\nabla \log p_t$, Assumption 2 is, to my understanding, contingent upon the loss function employed in training diffusion models. Consequently, it is not feasible to guarantee or even evaluate this assumption for diffusion models.
- I recommend redrawing Figure 1 in logarithmic scale to corroborate the theoretical findings.
- The proof appears to follow the approach outlined in [1]. I believe it is possible to enhance the sample complexity in the data dimension from $O(d^{3/2})$ to $O(d)$ by drawing techniques inspired by the state-of-the-art results presented in [2].
- I believe this paper lacks a comprehensive literature review. It fails to cite closely related empirical studies [3] and theoretical studies [4, 5], as well as the recent advancements in accelerating diffusion models, such as knowledge distillation [6], consistency models [7], adaptive stepsizes [8], parallel sampling [9], randomized midpoint [10], among others.

[1] Chen, Hongrui, Holden Lee, and Jianfeng Lu. “Improved analysis of score-based generative modeling: User-friendly bounds under minimal smoothness assumptions.” International Conference on Machine Learning. PMLR, 2023.

[2] Benton, Joe, et al. “Nearly d-linear convergence bounds for diffusion models via stochastic localization.” (2024).

[3] Dockhorn, Tim, Arash Vahdat, and Karsten Kreis. "Genie: Higher-order denoising diffusion solvers." Advances in Neural Information Processing Systems 35 (2022): 30150-30166.

[4] Wu, Yuchen, Yuxin Chen, and Yuting Wei. “Stochastic Runge-Kutta Methods: Provable Acceleration of Diffusion Models.” arXiv preprint arXiv:2410.04760 (2024).

[5] Li, Xuechen, et al. “Stochastic Runge-Kutta Accelerates Langevin Monte Carlo and Beyond.” Advances in Neural Information Processing Systems 32 (2019).

[6] Luhman, Eric, and Troy Luhman. “Knowledge Distillation in Iterative Generative Models for Improved Sampling Speed.” arXiv preprint arXiv:2101.02388 (2021).

[7] Mei, Song, and Yuchen Wu. “Deep Networks as Denoising Algorithms: Sample-Efficient Learning of Diffusion Models in High-Dimensional Graphical Models.” arXiv preprint arXiv:2309.11420 (2023).

[8] Jolicoeur-Martineau, Alexia, et al. “Gotta Go Fast When Generating Data with Score-Based Models.” arXiv preprint arXiv:2105.14080 (2021).

[9] Chen, Haoxuan, et al. “Accelerating Diffusion Models with Parallel Sampling: Inference at Sub-Linear Time Complexity.” arXiv preprint arXiv:2405.15986 (2024).

[10] Gupta, Shivam, Linda Cai, and Sitan Chen. "Faster Diffusion-based Sampling with Randomized Midpoints: Sequential and Parallel." arXiv preprint arXiv:2406.00924 (2024).

**Questions:**

- Assumptions 3 and 4 are both bounds for the third-order derivative of $\log p_t$. However, I firmly believe that temporal derivatives can be represented as spatial derivatives, thereby revealing fundamental properties of the data distribution, as shown in Equation (22) in [2]. Could you please clarify why Assumptions 3 and 4 are considered separate?
- If Assumption 2 is replaced with the corresponding assumption from [1], is the result for SDE-DPM-2 still valid? Is there any method to ensure the validity of this assumption during the training process?

---

> ### Author Response · Authors · 2024-11-21
>
> #  Response to Weakness:
> - **[assumptions]**: As discussed under Assumption 2, experimental work by Meng et al. (2021) designed a loss function that approximates both the score and its derivative, demonstrating that solvers using the estimated score and score derivative can achieve fast convergence. This suggests that the score's derivative can be well-approximated. Additionally, experiments by Meng et al. (2021) on Gaussian mixture models have shown that this method can effectively learn more accurate the score derivatives.
>
>   Furthermore, while some studies have theoretically proven that $L^2$ accuracy in score estimation is achievable when the target distribution is a simple Gaussian mixture (e.g., symmetric bimodal Gaussian mixtures), the theoretical understanding of the learnability of scores and second-order scores for more complex distributions remains limited. This is an area that merits further exploration.
>
> - **[ figure 1]**:  We have redrawn Figure 1 with a logarithmic scale to better illustrate the theoretical convergence rates of the different methods. The revised figure clearly shows that SDE-DPM-2 achieves a faster convergence rate than SDE-DPM and RK-2. Moreover, we acknowledge a gap between existing theoretical analyses and empirical observations, as experiments indicate that the KL divergence for each method decreases more rapidly than predicted by theoretical bounds. This discrepancy suggests potential directions for future research aimed at improving the convergence rates of these methods.
> - **[improve dimension dependence]**: Thank you for your insightful comment. We acknowledge that [2] achieves an improvement to sample complexity scaling from $ O(d^{3/2}) $ to $ O(d) $ by leveraging stochastic localization, which reduces the dependency of $ \mathbb{E}\|\nabla^2 \log q_t\|^2_F $ on the data dimension  $d$ . However, the solver we analyze in this work, SDE-DPM-2, is fundamentally different from [1] and [2]: SDE-DPM-2 relies on the first derivative of the score function during sampling.
>
>   In our analysis (as detailed in Appendix C.1), we employ a Taylor expansion to bound discretization error. Even when assuming a constant second derivative of the score, this approach introduces a dependency of  $O(d^{3/2})$. Therefore, at present, it is challenging to reduce the sample complexity's dimension dependency for SDE-DPM-2 to $ O(d) $.
> - **[literature review]**: We sincerely appreciate the reviewer’s suggestions and for highlighting several important studies related to diffusion models, which will enhance our theoretical understanding. Our primary focus is on the sampling complexity of diffusion model solvers, and we find that [4, 5, 9] are more closely related to our work. It is worth mentioning that [4] was published after our submission and introduces a novel stochastic RK-2 method. The assumptions in [5] may be a little stringent, requiring the target distribution to have a strongly convex potential, be four-times differentiable, and have Lipschitz continuous first three derivatives. Study [9] discusses parallel sampling methods, but each sampling step involves multiple evaluations of the score function, which may be computationally demanding. We have included discussions on these studies in our revised manuscript. The other studies suggested by the reviewer are mainly empirical or focus on ODE-based samplers, which we believe may not require extensive discussion in the related work section.

---

> ### Author Response · Authors · 2024-11-21
>
> #  Response to Questions:
> - **[assumptions 3 and 4]**: Unlike in [2], where the discretization error's derivative is directly bounded using expectations of $|\nabla \log p_t(x_t)|^2$ and $|\nabla^2 \log p_t(x_t)|^2$, our analysis focuses on the convergence of the SDE-DPM-2 method. This method introduces an additional term—the first derivative of the score function—into the discretization error. The presence of this extra term complicates the direct bounding of the error’s derivative as done in [2].
>
>   To address this, we decompose the discretization error into two components: temporal and spatial. We bound these components independently using Assumptions 3 (for the partial derivative with respect to $t$) and Assumption 4 (for the partial derivative with respect to $x_t$). We acknowledge that directly bounding the derivative of the discretization error in the same way as [2] may not be feasible due to the additional term in the SDE-DPM-2 discretization error, and we plan to explore this further in future work.
> - **[if assumption 2 is replaced]**: Due to the nature of our proof technique, Assumption 2 is necessary for the theoretical validation. However, it is worth noting that in our experimental comparisons between SDE-DPM-2 and SDE-DPM, we used commonly available pre-trained models (e.g., those trained for DDPM). Our results indicated that SDE-DPM-2 outperformed SDE-DPM even without specifically training to satisfy Assumption 2.
>
>   Similar to the common training objective of Denoising Score Matching (DSM), Assumption 2 can be validated in cases where the ground truth score and its derivative are explicitly computable, such as for Gaussian mixture distributions. This validation was demonstrated in the experiments by Meng et al. (2021). However, for more general distributions, neither Assumption 2 nor the assumption from [1] can be empirically validated during training at present.
>
> Chenlin Meng et al. "Estimating High Order Gradients of the Data Distribution by Denoising." (2021).

---

> > ### Comment · Reviewer_xHXz · 2024-11-25
> >
> > Thank you for your responses and clarifications. While I acknowledge its relevance, this work lacks the development of novel techniques, relaxation of assumptions, presentation of state-of-the-art results, and positioning within the existing literature, as also noted by Reviewer xXao. Thus, I retain my original score.

---

> > > ### Author Response · Authors · 2024-11-26
> > >
> > > Thank you for your feedback. In our revised manuscript, we have included state-of-the-art results [1, 2, 3, 4, 5, 6] to provide a comprehensive overview of the existing literature. Additionally, we have emphasized that our primary contribution lies in providing the **first convergence analysis of the SDE-DPM-2 method**, thereby addressing a theoretical gap.
> > >
> > > We would be happy to incorporate any additional state-of-the-art results if there are others we might have overlooked. Your insights and suggestions are greatly appreciated.
> > >
> > > References:
> > >
> > > [1] Daniel Zhengyu Huang, Jiaoyang Huang, and Zhengjiang Lin. Convergence analysis of probability flow ODE for score-based generative models. (2024)
> > >
> > > [2] Sitan Chen, Sinho Chewi, Holden Lee, Yuanzhi Li, Jianfeng Lu, and Adil Salim. The probability flow ODE is provably fast. (2024)
> > >
> > > [3] Gen Li, Yu Huang, Timofey Efimov, Yuting Wei, Yuejie Chi, and Yuxin Chen. Accelerating convergence of score-based diffusion models, provably. (2024)
> > >
> > > [4] Yuchen Wu, Yuxin Chen, and Yuting Wei. Stochastic Runge-Kutta methods: Provable acceleration of diffusion models. (2024)
> > >
> > > [5] Haoxuan Chen, Yinuo Ren, Lexing Ying, and Grant M. Rotskoff. Accelerating diffusion models with parallel sampling: Inference at sub-linear time complexity. (2024)
> > >
> > > [6] Xunpeng Huang, Difan Zou, Hanze Dong, Yi Zhang, Yi-An Ma, and Tong Zhang. Reverse transition kernel: A flexible framework to accelerate diffusion inference. (2024)

---

### Official Review · Reviewer_xXao · 2024-11-03

**Soundness:** 2
**Presentation:** 3
**Contribution:** 1
**Rating:** 3
**Confidence:** 5

**Summary:**

This paper analyzes the convergence of the higher-order discretization method (SDE-DPM-2). Under some smoothness condition as well as score estimation error and high oder estimation error, a sampling complexity at the order of O(1/epsilon) is established to ensure the KL divergence smaller than epsilon^2. In comparison, the complexity of second-order Runge–Kutta method (RK-2) scales as O(1/epsilon^2).

**Strengths:**

This paper analyzes the convergence of the higher-order discretization method (SDE-DPM-2). Under some smoothness condition as well as score estimation error and high oder estimation error, a sampling complexity at the order of O(1/epsilon) is established to ensure the KL divergence smaller than epsilon^2. In comparison, the complexity of second-order Runge–Kutta method (RK-2) scales as O(1/epsilon^2).

**Weaknesses:**

Although the following paper is posted after your submission, there maybe exist some conflict messages between your paper and this work: you said that RK-2 is less efficient, while this work claimed that RK-2 is provably fast.
Wu, Y., Chen, Y., and Wei, Y. Stochastic runge-kutta methods: Provable acceleration of diffusion models.

Li et al. (2024) also provided a sampling complexity of O(1/epsilon) under KL divergence and a better complexity of O(1/sqrt(epsilon)) for TV, which may reduce the theoretical contribution of this work and was not discussed here.

There exists some other convergence analysis for high-order sampling of diffusion models. It seems that their rates are better than yours, but such comparisons are missed here.
Huang, D. Z., Huang, J., and Lin, Z. Convergence analysis of probability flow ODE for score-based generative models.
Huang, X., Zou, D., Dong, H., Zhang, Y., Ma, Y.-A., and Zhang, T. Reverse transition kernel: A flexible framework to accelerate diffusion inference.

**Questions:**

No question.

---

> ### Author Response · Authors · 2024-11-21
>
> #  Response to Weakness
> - Wu et al. (2024) introduced a novel stochastic RK-2 method that differs from the stochastic RK-2 algorithm analyzed in our paper (Equations (9-10)). Our claim regarding the lower efficiency of RK-2 is based on its direct discretization of the linear component of the drift term in the SDE, which results in a higher error compared to the SDE-DPM-2 method. In contrast, Wu et al. (2024) developed a new stochastic RK-2 method that effectively incorporates the linear term of the drift, making it more efficient than the RK-2 method discussed in our work. We have also demonstrated that the Rk-2 analyzed in our paper is less efficient than SDE-DPM-2, as shown in Figure 1. We have clarified this distinction in the revised manuscript to avoid any potential confusion.
>
> - As discussed in Appendix A, our focus is on the existing SDE-DPM-2 method [Lu et al., 2022] and its theoretical significance. While Li et al. (2024) designed a different sampler and provided sampling complexities of $O(d^3/\sqrt{\epsilon})$ to achieve $\tilde{O}(\epsilon)$ in KL divergence and $O(d^3/\epsilon)$ to achieve $\tilde{O}(\epsilon)$ in TV, our contribution lies in analyzing the SDE-DPM-2 method.  SDE-DPM-2 has been experimentally validated for strong performance but lacked comprehensive theoretical analysis—a gap that our work fills. Our analysis demonstrates that SDE-DPM-2 achieves comparable complexities, specifically  $O(d^{1.5}/\sqrt{\epsilon})$ in KL divergence and $O(d^{1.5}/\epsilon)$ in TV, matching Li et al. (2024) in terms of $\epsilon$ dependence but with improved dependence on the data dimension $ d $.
>
>   Furthermore, the SDE-DPM-2 method is empirically more efficient: while both algorithms in Wu et al. (2024) and Li et al. (2024) achieve FID scores around 100 (Figure 2 in Wu et al., 2024), SDE-DPM-2 achieves an FID score of around 18 when sampling from the CIFAR-10 dataset with 20 steps. We have discussed this comparison in more detail in the revised manuscript to provide a clearer perspective on the theoretical contributions of our work.
> - Thank you for pointing out the comparison with these works. As we noted in our introduction, the first referenced work primarily provides convergence analysis for an ODE-based solver, whereas our focus is on SDE-based solvers. The second referenced work introduces a novel algorithm, RTK-ULD, and provides a sampling complexity of $\tilde{O}(L^2 d^{1/2} / \epsilon)$ to achieve $\tilde{O}(\epsilon)$ accuracy in total variation distance. It is important to emphasize that our analysis centers on existing algorithms, specifically SDE-DPM-2. While the second work reports a better dependence on the dimension $d$, our results maintain consistency regarding the $\epsilon$ dependence. Notably, the second paper includes dependencies on the Lipschitz constant, which is often associated with the data dimension, suggesting that their dimensional dependence might still be incomplete.

---

> > ### Comment · Reviewer_xXao · 2024-11-26
> >
> > Thanks for the response. I think I can treat this paper as a theory paper and provide review comments, as it didn't provide new method/idea that can achieve better performance or even potential. Then I think the authors should show the superiority of their analysis over other theoretical works that also studied the acceleration of diffusion models instead of just saying that they are irrelevant.
> >
> > Moreover, I think it is unfair to say that the modification in this paper is practical, while others' modification are not. For example, the stepsize {t_k} doesn't correspond to practical linear schedule in Lu et al 2022b, and they use x predictor for SDE-DPM-Solver++, while here score function is considered. Moreover, the assumption regarding the score estimation error is also different from practice, where eps predictor is trained and the errors for eps predictor and score function are very different for the steps close to the data distribution. Finally, it is unconvinced to say this is a practical acceleration algorithm which needs 20-100 steps for sampling task on CIFAR-10. In comparison, DPM-Solver++ and UniPC (Figure 2a) need only 5-8 steps to make FID achieve 5.
> >
> > Hence, I will keep my score.
> >
> > Wenliang Zhao, Lujia Bai, Yongming Rao, Jie Zhou, Jiwen Lu, UniPC- A Unified Predictor-Corrector Framework for Fast Sampling of Diffusion Models.

---

> > > ### Author Response · Authors · 2024-11-26
> > >
> > > Thank you for your further comments. We would like to clarify the practicality of our analysis and address the points you raised.
> > >
> > >   **Choice of Step Size**:
> > >   While Lu et al. (2022b) adopt a partial linear schedule for the step sizes $t_k$
> > >   , our analysis also supports such schedules in addition to the uniform step size commonly used in theoretical works.
> > >
> > >   **Predictor Choice**:
> > >   Regarding the predictor choice, we note that the use of the x-predictor in Lu et al. and the score function in our work are theoretically equivalent, as $x_\theta(x)=\frac{x+s_\theta}{\sigma_t}$. The decision to use the score function in our analysis aligns with the majority of existing theoretical works in the literature, maintaining consistency with the common approach for theoretical analysis.
> > >
> > >   **Epsilon-Predictor and Score Function**:
> > >   For the $\epsilon$-predictor commonly used in practical applications, Diederik et al. (Appendix D.1) show that the $\epsilon$-training objective is a reweighted form of denoising score matching, with similar theoretical guarantees in terms of estimation accuracy. Therefore, our assumptions regarding score estimation error are consistent with the widely accepted theoretical practices in the field.
> > >
> > >   **Performance on CIFAR-10**:
> > >   Finally, regarding the performance on CIFAR-10, we would like to further clarify the distinction between SDE-DPM-2 and ODE-based solvers such as DPM-Solver++ and UniPC. While DPM-Solver++ and UniPC, as ODE-based solvers, achieve lower FID values with fewer sampling steps, they may sacrifice image diversity and quality [Guo et al.; Shen et al.]. In contrast, SDE-based solvers like SDE-DPM-2 are capable of producing higher-quality and more diverse images [Lu et al.]. We provide results in Table 1 to demonstrate the significant acceleration achieved by SDE-DPM-2 compared to SDE-DPM. Based on both our theoretical analysis and experimental results, we argue that SDE-DPM-2 is a practical acceleration algorithm within the SDE-based framework.
> > >
> > >   We hope these clarifications address your concerns, and we thank you again for your valuable feedback. If you have any further questions or suggestions, we would be happy to address them.
> > >
> > >
> > >     Lu, Cheng, et al. "Dpm-solver++: Fast solver for guided sampling of diffusion probabilistic models." arXiv preprint arXiv:2211.01095 (2022).
> > >
> > >     Nie, Shen, et al. "The blessing of randomness: Sde beats ode in general diffusion-based image editing." (2023).
> > >
> > >     Kingma, Diederik, and Ruiqi Gao. "Understanding diffusion objectives as the elbo with simple data augmentation." (2024).
> > >
> > >     Guo, Hanzhong, et al. "Gaussian mixture solvers for diffusion models." (2024).

---

### Official Review · Reviewer_XcvQ · 2024-11-03

**Soundness:** 3
**Presentation:** 2
**Contribution:** 3
**Rating:** 6
**Confidence:** 4

**Summary:**

This paper studies the convergence properties of score-based diffusion models with a second-order discretization scheme called SDE-DPM-2, which improves the complexity over a first order exponential intergation scheme. Interestingly the result for SDE-DPM-2 is also stronger than the more widely used RK-2 scheme.

**Strengths:**

- The authors address a question of significant interest in the diffusion model literature, namely which discretization schemes are most sample efficient at inference time.
- The paper gives some additional theoretical support to the observation that higher order schemes can be important for sample complexity and differentiates between subtleties, such as the additional approximation in the linear term of the SDE.
- Experiments show a modest improvement of CIFAR-10 FID with small numbers of sampling steps and improved convergence with discretization fineness using SDE-DPM-2 over RK-2.

**Weaknesses:**

- There is no comparison of the computational cost of RK-2 vs DPM-SDE-2 vs EI
- I felt the authors should have more clearly delineated their contributions relative to Chen 2023, which they follow closely.
- A number of the assumptions are quite strong. For example, the expectation of the second time derivative of the score is assumed to have a magnitude upper bounded by some time-independent constant. In practice, it is often the case that the score changes in quite a singular fashion near $t=0$.

**Questions:**

- I think the authors might have a mistake in assumption 4, because I don't see them using the operator $\nabla^3$ anywhere.
- Can the authors add error bars to the table of the FID scores? At present, I don't feel that these illustrate their point particularly well.

---

> ### Author Response · Authors · 2024-11-21
>
> # Response to Weakness:
> - **[computational cost]**:  The primary factor influencing the computational cost of SGMs is the evaluation of the estimated score function, $s_theta$, typically represented by a large neural network. SDE-DPM-Solver++(2M) efficiently updates the derivative of the score function by reusing previously stored evaluations, thus matching the computational cost of a first-order method. In contrast, as shown in Equations (9-10), the RK-2 method requires evaluating the score function twice per time step, resulting in significantly higher computational cost. Consequently, both SDE-DPM-Solver++(2M) and EI have similar computational costs, which are notably lower than that of RK-2. We have clarified this comparison in the revised manuscript.
> - **[ comparision with Chen 2023]**: Chen et al. (2023) conducted a convergence analysis of the Exponential Integrator/SDE-DPM method. In contrast, our work focuses on the convergence analysis of the SDE-DPM-2 method, a second-order extension of the SDE-DPM method. We provide the first convergence analysis of SDE-DPM-2, demonstrating that it achieves a faster convergence rate compared to the original SDE-DPM method. We have compared the empirical performance of SDE-DPM-2 with SDE-DPM as demonstrated in Table 1 and Figure 1. We revised the manuscript to emphasize this distinction and clearly delineate our contributions from those of Chen et al. (2023).
> - **[assumptions]**: We have verified that under Gaussian mixture models, the score function does not exhibit highly singular behavior near $t=0$, confirming that the conditions outlined in Assumptions 3 and 4 hold. Since Gaussian mixtures are universal approximators for smooth probability density functions, these assumptions can be extended to a wide range of distributions that are effectively approximated by Gaussian mixtures. To clarify this, we have included a more detailed discussion in the revised manuscript. However, we acknowledge that the potential singular behavior of the score function for other types of distributions near $t=0$ remains less explored and presents an interesting direction for future research.
> #  Response to Questions:
> - **[$\nabla^3$]**:
>   We use the operator $\nabla^3$ in Assumption 4 to bound the discretization error term in Appendix C.1, the proof of Lemma 4.3. We have clarified this point in the revised manuscript to ensure that the use of the operator $\nabla^3$ is explicitly stated and justified.
> - **[error bar]**: We run each method multiple times and give the result in Table 1 of the revised manuscript, which further substantiates the improved experimental performance of SDE-DPM-2 over SDE-DPM.

---

> > ### Comment · Reviewer_XcvQ · 2024-11-25
> >
> > Thanks for these clarifications. I will maintain my score.

---

### Official Review · Reviewer_FNqr · 2024-11-03

**Soundness:** 4
**Presentation:** 2
**Contribution:** 3
**Rating:** 6
**Confidence:** 3

**Summary:**

Diffusion models (DMs) learn the score functions associated with a diffusion process, and use the learned scores to simulate an SDE corresponding to the backward process. While samples can be simulated by either an ODE or an SDE, SDE samplers are practically superior in terms of sample diversity and quality. This paper sets out to investigate second-order SDE solvers for the backward SDE, and concludes that second-order solver is preferable to the standard first-order discretization methods in terms of convergence with respect to the Kullback-Leibler divergence.

The paper mainly investigates two (approximate) second-order SDE solvers, SDE-DPM-2 and Runge-Kutta 2 methods, and compares the convergence results to first order SDE solvers such as EI. The paper presents theorems that suggest that SDE-DPM 2 is more preferable to RK-2 from the perspective of KL-divergence, mainly due to the added discretization error.

While the paper mainly focuses on the VP-DMs based on the Ornstein-Uhlenbeck forward process, the main result also applies to the variance-exploding forward process as well, shedding light on the applicability of solvers on other forward processes.

**Strengths:**

The paper presents convergence results of second-order SDE solvers for diffusion models, which is very relevant to current research in diffusion modeling given the empirical usefulness of SDE-based simulation of the backward process, and the open question of suitable discretization techniques in this context. The paper gives a theoretical foundation on the application of high-order SDE solvers in diffusion modeling, which motivates further research on suitable solvers for diffusion generative modeling.

- Within the scope of the paper, it presents a compelling argument in favor of SDE-DPM-2 over RK-2 or first-order discretization methods for the practical simulation of samples. I find the insight of "not second-order solvers are equal" overall interesting and helpful.
- The paper also illustrates that the convergence bounds empirically with Gaussian mixture examples.
- The theoretical results are quite general as they apply to both VP and VE diffusion models.

**Weaknesses:**

While I have an overall positive outlook on the paper, I think the paper's overall organization seems confusing: the paper presents the main theorems and some empirical results, then jumps back to a sketch of the proof and how the theory works for the VE-type diffusion models. In my opinion, presenting the paper as theorems on SDE-DPM-2 and RK-2, proof sketch, discussion on VE and then experiments seems like more logical progression of the narrative.

There are a number minor issues in terms of the paper's presentation. Here is a list I have found:
- Many discretization methods mentioned in the paper are known only as acronyms without mentioning what the acronyms are.
- The mentions of $x_k$ in assumption should be $x_{t_k}$ in equations such as the one in Assumption 2, eqs. 11 and 13.
- The use of partial derivatives w.r.t. $x_{t_k}$ seems confusing. I assume it means the Jacobian matrix. Perhaps the authors can explicitly denote a notation to describe the Jacobian matrix for clarity.
- While it is useful to see that second-order SDE solvers makes improvements empirically, Table 1 presents quite little added information other than a somewhat vague empirical confirmation that SDE-DPM-2 does have empirical value, which has already been demonstrated by Lu et al. (2022b).

**Questions:**

- The paper presents the KL convergence results about the VP- and VE-types of diffusions models separately. Could you briefly explain how different types of forward processes affect your proof?
- I find panel (b) of Figure 1 quite helpful as an illustration between theory and practice, but the paper also presents convergence results with respect to RK-2. What would the theoretical bounds of RK-2 look like on that graph?
- Are equations 11 and 13 identical? If so, why?

---

> ### Author Response · Authors · 2024-11-21
>
> # Response to Weakness:
>  - **[Structure]**: We agree that the structure you suggested is indeed more logical and easier to follow.  We have revised the manuscript accordingly to reflect this improved structure.
> - **[Acronyms for Discretization Methods]**: We appreciate your feedback and will make sure to define all acronyms for the discretization methods used in the paper to avoid any confusion for the readers.
>
>  - **[Notation for $x_k$]**: We will clearly point out that $x_k$ refers to $x_{t_k}$ in the manuscript to avoid any confusion.
>
>   - **[Partial Derivatives with Respect to $x_{t_k}$]**: We have clarified that the partial derivatives with respect to $x_{t_k}$ refer to the Jacobian matrix and explicitly defined this notation for clarity in the revised manuscript.
>
>   - **[Table 1 and Empirical Confirmation]**: While Lu et al. (2022b) demonstrated that SDE-DPM-2 generates better samples than SDE-DPM through image generation examples, our Table 1 provides additional support for this claim by comparing specific metrics (e.g., FID scores). This further substantiates the improved experimental performance of SDE-DPM-2 over SDE-DPM, offering a more detailed empirical validation of its effectiveness. We run each method multiple times and give the result in table 1.
> # Response to Questions:
> - **[VP and VE]**:  The main distinction between the two types of forward processes lies in the behavior of the initial error term. In the VP-type diffusion model, the initial error term decays exponentially fast, contributing minimally to the overall convergence. In contrast, for the VE-type diffusion model, as outlined in equation (17), the initial error term decays at a polynomial rate, specifically $1/T$, which becomes the dominant factor compared to the discretization error term $1/T^2$. Consequently, we conduct separate convergence analyses for the VP- and VE-type diffusion models.
> - **[Figure 1]**:  We have incorporated the empirical and theoretical convergence results for RK-2 into Figure 1, employing a logarithmic scale to depict the theoretical bounds of each method. Our analysis demonstrates that, while RK-2 and SDE-DPM show comparable empirical performance, both are less efficient than SDE-DPM-2. Moreover, we identified a gap between existing theoretical results and empirical observations, as the KL divergence for each method decreases more rapidly than the theoretical bounds. This may suggest potential directions for future research to further improve the convergence rate of these methods.
> - **[equation 11 and 13]**: Equations (11) and (13) are not identical. The drift terms in the two equations differ: Equation (11) contains $\hat{x}_{t_K}$ in the drift term, which is a constant within each time interval, whereas Equation (13) includes $x_t$ in the drift term, which is time-dependent. We clarified this distinction in the revised manuscript, as it represents the key difference between the SDE-DPM-2 and RK-2 methods.

---

### Author Response · Authors · 2024-11-21
**Revised Version**

We sincerely appreciate the reviewers' valuable suggestions and comments. We have carefully revised the manuscript accordingly. Please see the details in the updated manuscript.

---

### Meta-Review · Area_Chair_BNyH · 2024-12-11

**Metareview:**

This paper develops non-asymptotic convergence rates for second-order diffusion samplers such as SDE-DPM-2 and RK-2. However, the assumptions made in this paper are much stronger than comparable studies existing in the literature, and it has not adequately compared with existing literature, making it less significant and not ready for publication.

**Additional Comments On Reviewer Discussion:**

During the discussion, the authors have incorporated additional discussion with missing literature, however, the main issues in terms of strong assumptions and limited analysis novelty still stand.

---

### Decision · Program_Chairs · 2025-01-22

Reject